# A broadly protective therapeutic antibody against influenza B virus with two mechanisms of action

Ning Chai[1], Lee R. Swem[1,†], Summer Park[2], Gerald Nakamura[3], Nancy Chiang[3], Alberto Estevez[4], Rina Fong[4], Lynn Kamen[5], Elviza Kho[5], Mike Reichelt[6], Zhonghua Lin[2], Henry Chiu[7], Elizabeth Skippington[8], Zora Modrusan[9], Jeremy Stinson[9], Min Xu[2], Patrick Lupardus[4], Claudio Ciferri[4] & Man-Wah Tan[1]

Influenza B virus (IBV) causes annual influenza epidemics around the world. Here we use an *in vivo* plasmablast enrichment technique to isolate a human monoclonal antibody, 46B8 that neutralizes all IBVs tested *in vitro* and protects mice against lethal challenge of all IBVs tested when administered 72 h post infection. 46B8 demonstrates a superior therapeutic benefit over Tamiflu and has an additive antiviral effect in combination with Tamiflu. 46B8 binds to a conserved epitope in the vestigial esterase domain of hemagglutinin (HA) and blocks HA-mediated membrane fusion. After passage of the B/Brisbane/60/2008 virus in the presence of 46B8, we isolated three resistant clones, all harbouring the same mutation (Ser301Phe) in HA that abolishes 46B8 binding to HA at low pH. Interestingly, 46B8 is still able to protect mice against lethal challenge of the mutant viruses, possibly owing to its ability to mediate antibody-dependent cellular cytotoxicity (ADCC).

[1] Department of Infectious Diseases, Genentech, South San Francisco, California 94080, USA. [2] Department of Translational Immunology, Genentech, South San Francisco, California 94080, USA. [3] Department of Antibody Engineering, Genentech, South San Francisco, California 94080, USA. [4] Department of Structural Biology, Genentech, South San Francisco, California 94080, USA. [5] Department of BioAnalytical Sciences, Genentech, South San Francisco, California 94080, USA. [6] Department of Pathology, Genentech, South San Francisco, California 94080, USA. [7] Department of Biochemical and Cellular Pharmacology, Genentech, South San Francisco, California 94080, USA. [8] Department of Bioinformatics and Computational Biology, Genentech, South San Francisco, California 94080, USA. [9] Department of Molecular Biology, Genentech, South San Francisco, California 94080, USA. † Present address: Therapeutic Antibody Department, Achaogen, South San Francisco, California 94080, USA. Correspondence and requests for materials should be addressed to N.C. (email: ningc@gene.com) or to M.-W.T. (email: mant@gene.com).

Each year influenza virus is estimated to cause three to five million cases of severe illness and around half million deaths worldwide[1]. Influenza B virus (IBV) co-circulates with influenza A virus (IAV) to cause annual influenza epidemics and accounts for a significant part of the influenza burden. IBV contains two major surface proteins, hemagglutinin (HA) and neuraminidase. Based on antigenic and genetic variation of HA, IBV is classified into two distinct lineages, Victoria and Yamagata, which co-circulate globally[2]. During 2004–2013, IBV contributed to 1–56% of all circulating influenza virus strains and 1–52% of influenza-associated paediatric mortality[3]. IBV infections appear to be more severe among paediatric patients. For example, IBV accounted for 38% of all influenza-associated paediatric deaths despite constituting only 26% of all circulating influenza viruses during the 2010–2011 season[4]. Vaccination is the primary means for the prevention of IBV, whereas neuraminidase inhibitors (NAIs) are early treatment and prophylaxis options. Among children, recent clinical studies showed that NAIs are less effective in treating IBV than IAV[5,6]. Alternative anti-IBV treatment with superior efficacy is highly needed.

Antiviral antibodies of the IgG class are often bi-functional[7,8]. Binding of the fragment antigen-binding (Fab) domain of IgG to viral epitopes can directly neutralize viral infectivity. In addition, upon Fab-mediated binding to viral antigens on the surface of infected cells, the fragment crystallisable (Fc) domain of IgG can provide protective activity in vivo by engaging host effector cells to kill virus-infected cells through antibody-dependent cellular cytotoxicity (ADCC) or complement-dependent cytotoxicity (CDC). ADCC is initiated when the Fc regions bind to activating Fcγ receptors on effector cells, such as natural killer (NK) cell, leading to a cascade of signalling events within the effector cells to trigger lysis of the infected cells. CDC, on the other hand, is induced by the deposition of complement on the surface of opsonized infected cells leading to the formation of membrane attack complex to lyse the infected cells.

Recently, human monoclonal IgGs that target the HA protein and protect mice from lethal IBV infections have been described[9,10]. HA mediates three important processes during the viral life cycle: (1) mediates viral attachment to host cells through interactions of the receptor-binding pocket (RBP) in the head domain with the sialic acid receptors, (2) undergoes dramatic conformational changes in the stalk region upon endosomal acidification to fuse the viral and endosomal membranes and (3) facilitates egress of viral particles from infected cells[11,12]. In general, neutralizing antibodies binding to the RBP block viral attachment, whereas those targeting the HA stalk block membrane fusion, as exemplified by the broadly neutralising monoclonal antibodies (mAbs) CR8033 and CR9114, respectively[9]. Some head-binding mAbs can also inhibit viral egress, similar to the action of NAIs such as oseltamivir phosphate (Tamiflu)[9,11]. In addition, some stalk-binding mAbs are able to prevent HA activation, the cleavage of HA0 by host serine proteases into two-disulfide-linked subunits HA1 and HA2 (refs 13,14). Another broadly neutralising anti-IBV mAb, CR8071 binds to the vestigial esterase domain in HA head, representing a new class of neutralising epitopes on IBV HA that are highly conserved[9]. CR8071 neutralized IBV in vitro by inhibiting viral egress and protected mice from lethal infection when administered prophylactically[9]. CR8071 is also able to induce ADCC in vitro[15]. Yet another antibody, 5A7 binds to the C terminus of HA1 in the stalk and partially protected mice (20% or 60%) from lethal IBV infection when administered 72 h post infection[10]. Intriguingly, 5A7 was able to block both viral attachment and membrane fusion[10].

Despite the progress in isolating broadly neutralising anti-IBV mAbs, the neutralisation spectrum and therapeutic efficacy of these mAbs are not optimal. Also, the resistance mechanisms against these antibodies and whether these mAbs could elicit ADCC or CDC have not been thoroughly studied. Here we report the discovery and characterization of a human IgG1 antibody, 46B8 that neutralizes all IBV strains tested in vitro and protects mice against the lethal challenge of a panel of IBVs with improved therapeutic benefits compared with Tamiflu.

## Results

**Isolation of a broadly neutralising human mAb against IBV.** We used a previously described in vivo antigen-specific plasma-blast enrichment technique to isolate broadly neutralising human anti-IBV mAbs from blood donors 7 days post vaccination[16,17]. In brief, peripheral blood mononuclear cells were isolated from human blood or leukopak and mixed with IBV HA protein to activate antigen-specific cells prior to intrasplenic transplantation into severe combined immunodeficiency (SCID) mice for rapid expansion and enrichment of human plasmablasts. Individual IBV HA-specific plasmablasts were isolated from splenic cells of the SCID mice by flow cytometry and subjected to IgG cloning followed by Enzyme-linked immunosorbent assay (ELISA) screening of the cloned mAbs. In order to select for broadly reactive mAbs that bind to highly conserved epitopes in HA, we used HA variants from both the Yamagata and Victoria lineages as well as the ancestral strains of IBV for the enrichment and sorting of IBV-specific plasmablasts and ELISA screening of cloned IgGs (Supplementary Fig. 1a). From 2018 IBV HA-binding plasmablasts, we identified 99 mAbs (IgG1) that bound to all three HAs used for screening (Supplementary Fig. 1b). Among these, three mAbs, 46B8, 34B5 and 33F8, neutralized representative IBVs from both lineages and the ancestral strains (Supplementary Fig. 2). However, 34B5 and 33F8 showed reduced efficacies against the B/Wisconsin/1/2010 virus. By contrast, 46B8 was able to neutralize all eleven IBV strains tested that were associated with human infections spanning >70 years, with half-maximal inhibitory concentration ($IC_{50}$) values ranging from 0.58 to 0.95 nM (Table 1), consistent with its ability to bind to a panel of IBV HAs (Supplementary Fig. 3). The germlines used by 46B8 are IGHV5-51*01 and IGKV2-28*01, different from those used by other broadly neutralising anti-IBV mAbs[9,10].

**46B8 blocks membrane fusion and induces ADCC in vitro.** We next studied the neutralisation mechanisms of 46B8. First, 46B8 did not show hemagglutination-inhibition (HI) activity even at

**Table 1 | In vitro neutralisation of influenza B viruses by 46B8.**

| Influenza B Strain | Lineage | $IC_{50}$ (nM) | 95% CI (nM) |
|---|---|---|---|
| B/Phuket/3073/2013 | Yamagata | 0.73 | 0.65–0.82 |
| B/Massachusetts/2/2012 | Yamagata | 0.69 | 0.61–0.78 |
| B/Wisconsin/1/2010 | Yamagata | 0.64 | 0.53–0.76 |
| B/Bangladesh/3333/2007 | Yamagata | 0.75 | 0.62–0.92 |
| B/Victoria/504/2000 | Yamagata | 0.67 | 0.62–0.73 |
| B/Brisbane/60/2008 | Victoria | 0.86 | 0.72–1.04 |
| B/Malaysia/2506/2004 | Victoria | 0.58 | 0.36–0.91 |
| B/Russia/1/1969 | Ancestral | 0.73 | 0.61–0.86 |
| B/Massachusetts/3/1966 | Ancestral | 0.80 | 0.65–0.98 |
| B/Maryland/1/1959 | Ancestral | 0.95 | 0.71–1.26 |
| B/Lee/10/1940 | Ancestral | 0.68 | 0.55–0.85 |

CI, confidence interval; $IC_{50}$, half-maximal inhibitory concentration.
Neutralisation was tested in a plaque reduction assay.

125 µg ml $^{-1}$ (Supplementary Fig. 4a), indicating that it does not block viral attachment to cell-surface receptors. By contrast, mAb 34B5 demonstrated robust HI activity against B/Victoria/504/2000 (Supplementary Fig. 4a), possibly binding to the RBP at the top of the HA head. 34B5 showed decreased HI activity for B/Wisconsin/1/2010, consistent with its reduced neutralisation efficacy against this virus. When an I196T substitution in the 190-helix of RBP (numbering excludes the signal sequence)[18] was introduced into the B/Wisconsin/1/2010 HA, it conferred susceptibility to 34B5 binding and sensitivity to 34B5 neutralisation of pseudotype virus (Supplementary Fig. 5a and 5b), supporting a binding site in the RBP. Since 46B8 does not block viral attachment to receptors, we next tested whether it possesses the neutralisation mechanisms of stalk-binding mAbs, such as inhibiting HA0 maturation or membrane fusion. Western blot analysis showed that 46B8 did not block trypsin-mediated HA0 activation as the HA0 protein was rapidly cleaved by trypsin in the presence of 46B8 (Supplementary Fig. 4b).

The ability of 46B8 to block HA-mediated membrane fusion was assessed in a cell–cell fusion assay (Fig. 1a). Cells expressing IBV HA fused to form giant syncytia at low pH; this was observed in the presence of a control IgG or 34B5, which likely binds to the RBP of HA. In sharp contrast, 46B8 completely blocked fusion. To provide further evidence that 46B8 blocks membrane fusion

by preventing low pH-induced conformational changes in HA, we performed an established HA conformational change assay using flow cytometry[13], in which cell surface HA undergoes conformational changes at low pH and subsequent dithiothreitol (DTT) treatment causes dissociation of the HA1 subunit from the membrane-anchored HA2 subunit (Fig. 1b, upper panel). We used 34B5 binding as an indication for the presence of the HA1 subunit as it likely binds to the RBP in HA head. As expected, HA-expressing cells pre-incubated with the control IgG lost 34B5 binding upon low pH and DTT treatments (Fig. 1b, lower panel). By contrast, pre-incubation with 46B8 restored 34B5 binding, indicating preservation of the prefusion HA conformation even at low pH (Fig. 1b). These results suggest that 46B8 blocks IBV infection in a neutralisation assay by preventing low pH-induced conformational changes in HA during the membrane fusion step (Supplementary Fig. 2).

It has been shown recently for broadly neutralising anti-IAV mAbs against either the head or stalk domain of HA that the Fc function can also provide protection in vivo through Fc receptor engagement and the induction of ADCC[8]. To determine whether 46B8 could induce ADCC, we performed two complementary assays to detect ADCC in vitro: (a) NK cell activation, by measuring the surface expression level of the NK cell activation marker CD107a (LAMP-1), which tightly correlates with cytokine

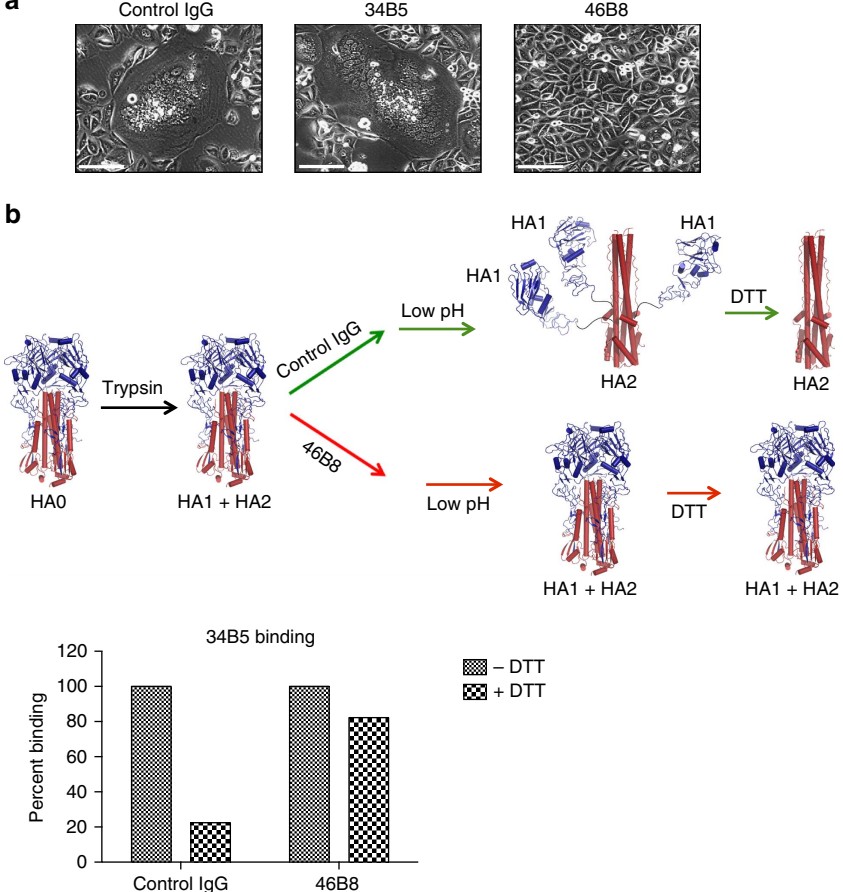

**Figure 1 | 46B8 blocks low pH-induced membrane fusion and conformational changes of HA.** (**a**) Hela cells expressing the B/Wisconsin/1/2010 HAs were treated with trypsin to activate HA0 and then incubated with either 46B8 or a control IgG before pH drop to 4.8 to induce cell–cell fusion. After overnight culture, representative images were obtained under a phase contrast microscope. Scale bar is 200 µm. (**b**) 293 T cells expressing the B/Victoria/504/2000 HA were treated with trypsin to activate HA0 and incubated with either 46B8 or a control IgG before pH drop to 4.8 to induce HA conformational changes. Cells were then treated with DTT (+DTT) or PBS (−DTT) and tested for 34B5 binding by flow cytometry. The mean fluorescence intensities were normalized to the PBS-treated cells in each group and the percent of binding is shown. Possible HA conformations at each stage are depicted above the binding data. HA1 is in blue and HA2 is in red.

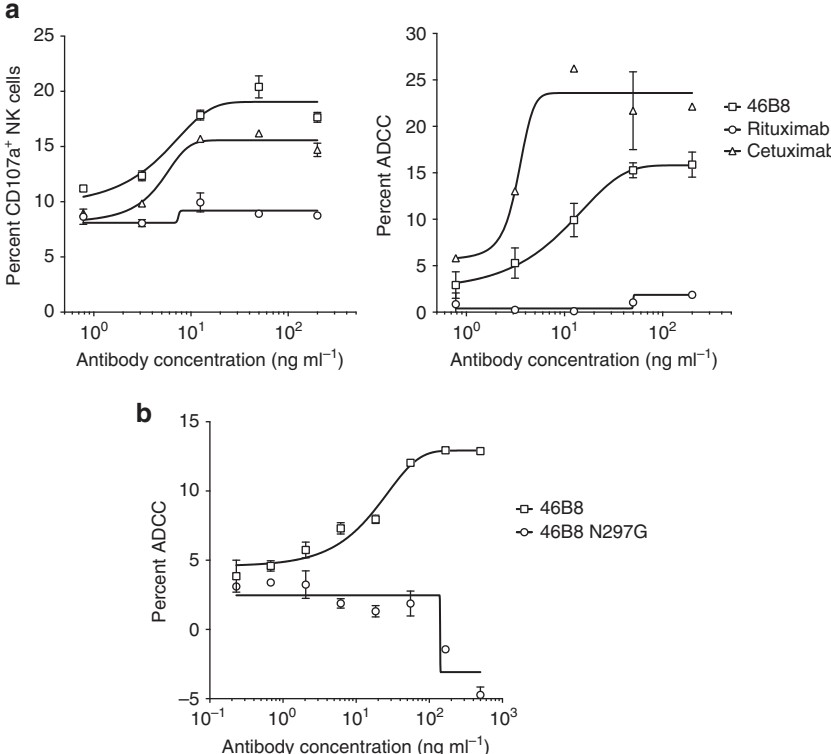

**Figure 2 | 46B8 induces ADCC *in vitro*.** (**a**) A549 cells infected with B/Brisbane/60/2008 were labelled with 46B8, Rituximab (negative control) or Cetuximab (positive control) prior to incubation with NK cells. NK cell activation (left panel) and target cell lysis (right panel) were presented as the frequency of CD107a$^+$ cells and the percent of LDH release, respectively. (**b**) A549 cells infected with B/Brisbane/60/2008 were labelled with 46B8 or 46B8 N297G prior to incubation with NK cells. Target cell lysis was presented as the percent of LDH release. The assays were done in duplicate. Results are representative of three independent experiments. (**a**,**b**: mean and s.e.m.)

production and cytotoxicity by NK cells[19], and (b) target cell lysis, by measuring the release of lactate dehydrogenase (LDH) from lysed cells. Target A549 cells were infected with B/Brisbane/60/2008 and subsequently coated with different mAbs prior to mixing with freshly isolated NK cells. Rituximab, a negative control mAb that mediates ADCC via CD20 (ref. 20) (not expressed on A549 cells), did not induce CD107a expression (Fig. 2a, left panel) or target cell lysis (Fig. 2a, right panel). By contrast, 46B8 induced both CD107a expression and target cell lysis to levels comparable to those achieved by Cetuximab, a positive control mAb that mediates ADCC via EGFR[21] (expressed on A549 cells). A 46B8 variant with an N297G mutation in Fc (46B8 N297G) that abolishes FcγR binding[22] was not able to induce ADCC, validating the involvement of Fc-FcγR interaction in 46B8-mediated ADCC (Fig. 2b).

To determine whether 46B8 could also induce CDC, virus-infected A549 cells were coated with 46B8 or Rituximab (negative control) followed by incubation with human serum complement. As a positive control, WIL2-S cells (expressing CD20) were coated with Rituximab prior to complement exposure[23]. After cell lysis, we used a luminescent substrate to detect ATP in the remaining live cells that were not lysed by CDC (Supplementary Fig. 6). As expected, Rituximab caused lysis of the WIL2-S cells but not the A549 cells. 46B8 did not induce significant CDC on A549 cells infected with the B/Brisbane/60/2008 virus.

**Structural characterization of the 46B8 epitope on IBV HA.** To understand the structural basis for virus neutralisation, we generated a 3D reconstruction of the ectodomain of B/Victoria/504/2000 HA in complex with a 46B8 Fab using negative stain electron microscopy (EM) and single particle analysis (Supplementary Fig. 7). Crystal structures of the B/Yamanashi/166/1998 HA (PDB ID 4M40) and a 46B8 Fab homology model were used as atomic models for fitting into the EM density. Each HA trimer was bound by three 46B8 Fabs (Fig. 3a, right panel). Interestingly, 46B8 binds to the vestigial esterase domain at the base of the HA head between the RBP and the stalk region, and at an angle tilting upward (Fig. 3a, left panel). In addition, the electron density covered by one 46B8 Fab appears to span two adjacent HA monomers (Fig. 3a, right panel). The binding site of 46B8 on HA is also consistent with its lack of HI activity and inability to block HA0 cleavage (Supplementary Fig. 8).

To determine the 46B8 epitope, we selected nine potential 46B8-contacting residues on B/Victoria/504/2000 HA based on the EM reconstruction (Fig. 3b, numbering includes the signal sequence) and made single point mutations that changed the size or charge of the individual residues. We expressed these HA mutants on 293 T cells and tested for 46B8 binding by flow cytometry. All nine HA mutants were expressed on the cell surface as they bound 34B5 at levels comparable to WT HA (Fig. 3c, top panel). At pH 7.0, three HA mutants, P302A, S300F and E306K, showed reduced binding to 46B8 whereas the remaining mutants bound 46B8 similarly to the WT HA (Fig. 3c, middle panel). As 46B8 does not block viral attachment to cells and is likely internalized into endosomes/lysosomes with IBV, we tested binding at pH 4.8, the average pH of lysosomes[24]. Seven of the nine HA mutants showed reduced 46B8 binding at low pH (Fig. 3c, bottom panel). S300F and P302A completely abolished binding whereas K53E, H55E and V105F decreased binding by more than 10-fold. E306K and T52A reduced binding moderately, whereas L73F and N74A bound 46B8 similarly to the WT HA. These results identified S300, P302, K53, H55, V105,

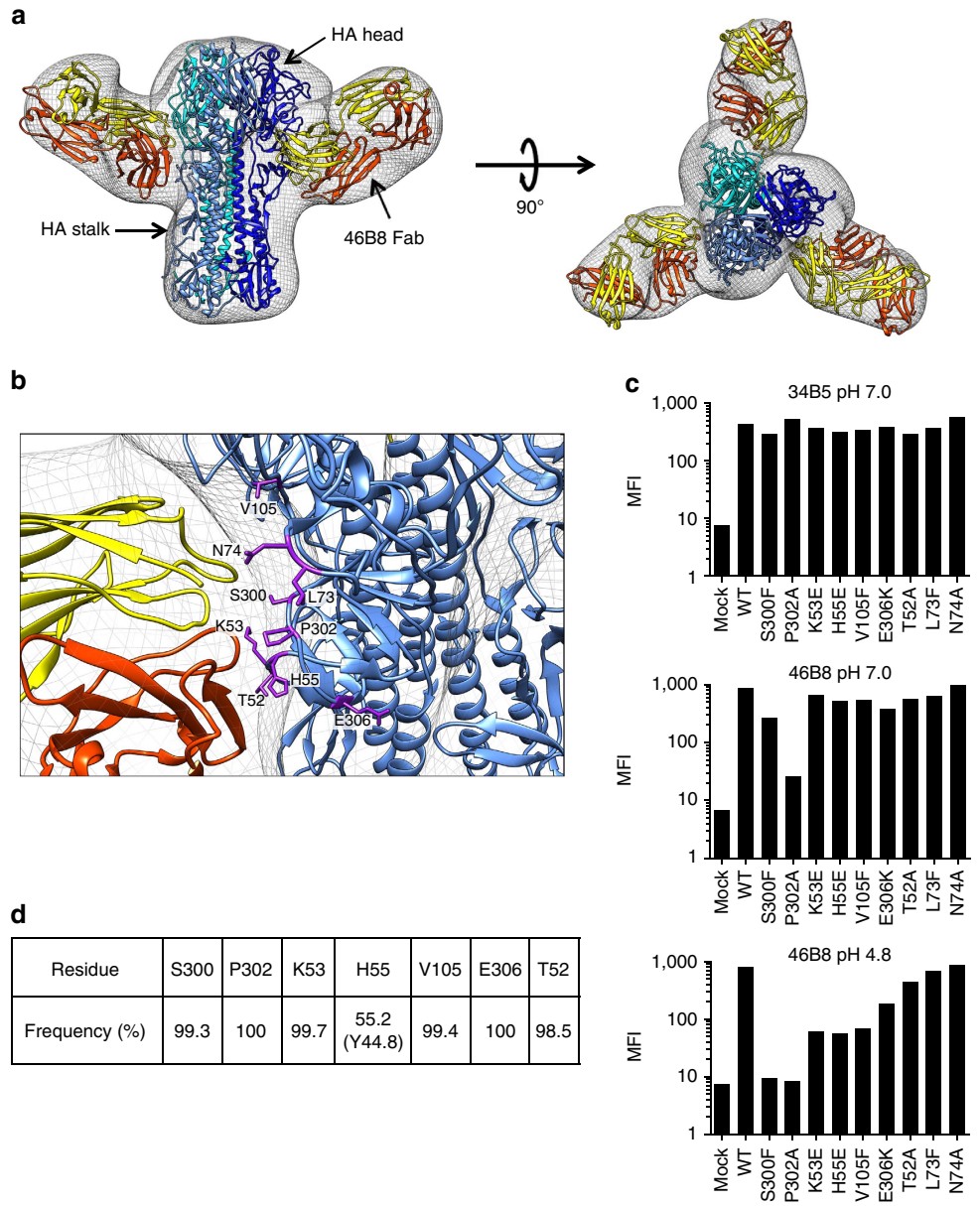

**Figure 3 | 46B8 binds to conserved residues in the vestigial esterase domain of HA.** (**a**) Negative stain EM reconstruction (grey mesh) of 46B8 Fabs in complex with the B/Victoria/504/2000 HA trimer. Side (left panel) and overhead (right panel) views show the fit of a generic Fab model and the crystal structure of an IBV HA (PDB ID 4M40) to the EM density. HA is in different shades of blue, the heavy chain of 46B8 Fab is in red and the light chain is in yellow. (**b**) A detailed view of the 46B8-HA contact surface. Potential 46B8-interacting residues on HA are shown as purple sticks and labelled (numbering includes the signal sequence). Colour code is the same as in **a**. (**c**) 293 T cells expressing the WT or mutant B/Victoria/504/2000 HAs were incubated with 34B5 (top panel) or 46B8 (middle and bottom panels) at pH 7.0 (top and middle panels) or 4.8 (bottom panel). The mean fluorescence intensities (MFI) from flow cytometry profiles are shown. Mock, mock transfected cells. (**d**) Frequency table of potential 46B8-interacting residues. A multiple sequence alignment of 12,790 human IBV HA amino-acid sequences was used to assess the genetic diversity and calculate the frequencies of potential 46B8-interacting residues.

E306 and T52 as potential 46B8 epitope residues. Six of these seven residues are highly conserved (>99% among all human IBV isolates) with the exception of H55 (Fig. 3d).

**Efficacy of 46B8 in mice**. Given the broad and robust *in vitro* neutralisation activity of 46B8 and its distinct mechanism from NAIs, we examined the efficacy of 46B8 alone or in combination with Tamiflu in various mouse infection models. We first tested 46B8 alone against five IBV isolates from both lineages and the ancestral strains spanning 1966–2010. Mice were challenged with

a minimum lethal intranasal dose of each virus and received either a single intravenous administration of 15 mg kg$^{-1}$ of 46B8 or a control IgG at 24, 48 or 72 h post infection. Control IgG-treated mice exhibited 100% mortality by day 9–12 with the exception of a single B/Massachusetts/3/1966-infected mouse (Fig. 4a). By contrast, 46B8 treatment at 24 and 48 h resulted in 100% protection, with the only exception of mice infected with B/Russia/1/1969 that showed a 60% protection when treatment commenced at 48 h. Even when given at 72 h post infection, 48B8 was highly efficacious resulting in 100% protection against the B/Brisbane/60/2008, B/Victoria/504/2000 and B/Massachusetts/3/

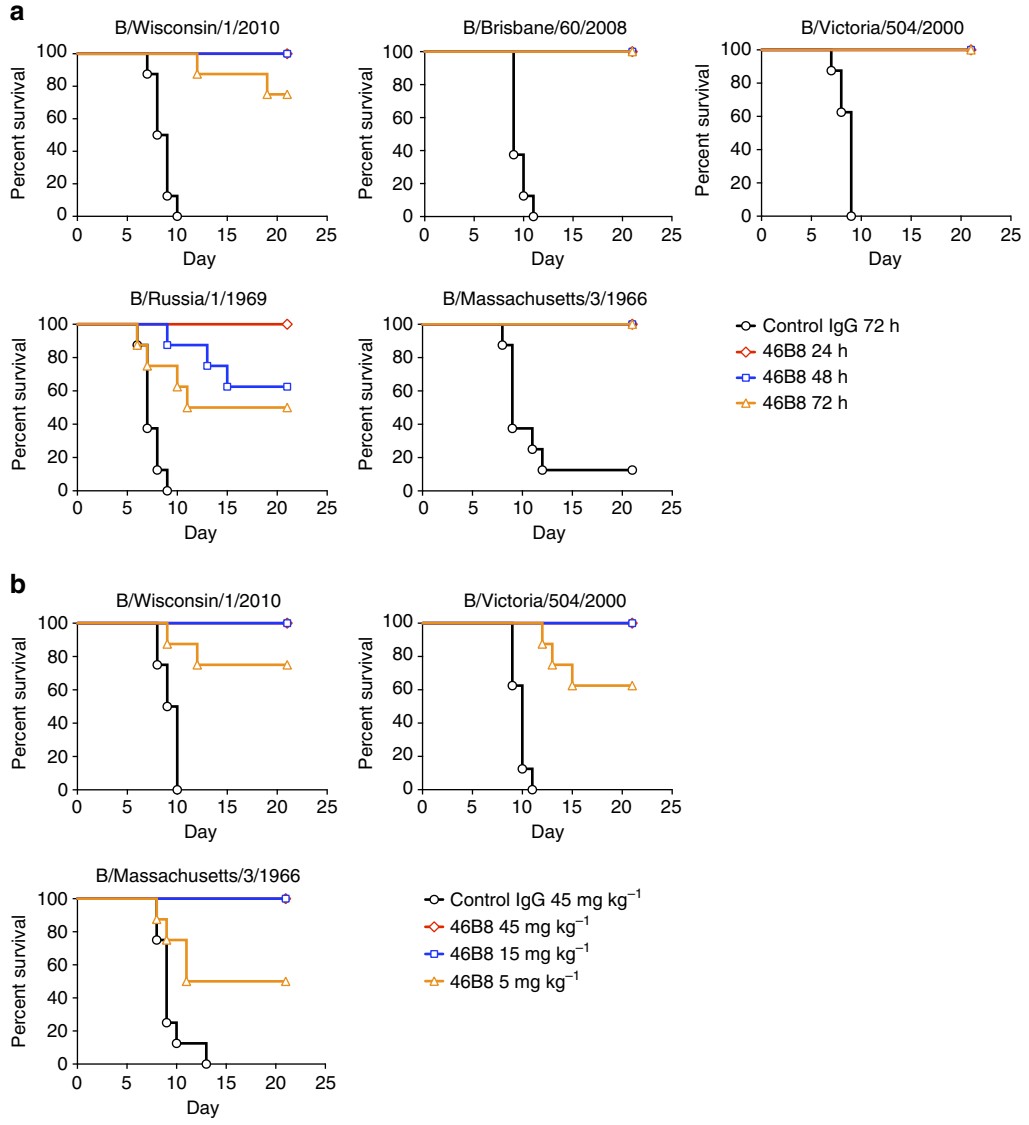

**Figure 4 | *In vivo* efficacy of 46B8 in influenza B mouse infection models. (a)** DBA/2 J mice were infected intranasally with a minimum lethal dose of B/Wisconsin/1/2010, B/Brisbane/60/2008, B/Victoria/504/2000, B/Russia/1/1969 or B/Massachusetts/3/1966. At 24, 48 or 72 h post infection, mice received a single treatment of 46B8 or a control IgG intravenously at 15 mg kg$^{-1}$. Survival curves are shown. Each group contains eight mice. Log-rank tests of all 46B8-treated groups versus the control in all infection models give $P < 0.05$. **(b)** DBA/2 J mice were infected intranasally with a minimum lethal dose of B/Wisconsin/1/2010, B/Victoria/504/2000 or B/Massachusetts/3/1966. At 72 h post infection, mice received a single treatment of 46B8 or a control IgG intravenously at 5, 15 or 45 mg kg$^{-1}$. Survival curves are shown. Each group contains eight mice. Log-rank tests of all 46B8 doses versus the control in all infection models give $P < 0.05$.

1966 strains, 75% protection against B/Wisconsin/1/2010 and 50% protection against B/Russia/1/1969. When treatment with 46B8 commenced at 72 h post infection, dose-dependent protection was observed (Fig. 4b). At both 45 and 15 mg kg$^{-1}$, 46B8 conferred 100% protection against all three IBV strains tested. Although efficacy was less pronounced at 5 mg kg$^{-1}$, 46B8 still provided significant protection as compared with 45 mg kg$^{-1}$ of the control IgG. Consistent with the survival benefit, 46B8-treated mice that survived the infection showed reduced body weight (BW) loss and were able to regain 100% of their initial pre-infection weight (Supplementary Fig. 9).

As 48B8 is efficacious beyond the optimal therapeutic window of Tamiflu (48 h)[25], we compared efficacies of 48B8 and Tamiflu in mice challenged with either a minimum lethal dose (Fig. 5a, left panels) or a high lethal dose (4 × the minimum dose, Fig. 5a, right panels) of B/Victoria/504/2000. At 72 h post infection, mice received either a single intravenous administration of 46B8

(45 mg kg$^{-1}$) or Tamiflu (100 mg kg$^{-1}$) orally twice a day for 5 consecutive days. All the control-treated mice died by day 9 (Fig. 5a, top panels). In the minimum lethal dose model, Tamiflu protected only 58% of the mice, whereas the single dose of 46B8 protected all treated mice (Fig. 5a, top left panel). The difference was more pronounced in the high lethal dose model where 75% of 46B8-treated mice survived, whereas all Tamiflu-treated mice died (Fig. 5a, top right panel). Also, mice treated with 46B8 showed reduced BW loss and regained their weight much faster than Tamiflu-treated mice that survived the infection (Fig. 5a, bottom panels). These results demonstrated superior therapeutic benefits of 46B8 over Tamiflu at 72 h post infection.

Despite the reduced efficacy of Tamiflu beyond its optimal therapeutic window, we anticipate that severe influenza patients will be given Tamiflu upon hospital admission. Therefore, we tested whether co-administration of 46B8 and Tamiflu improves efficacy over either treatment alone in the high lethal dose

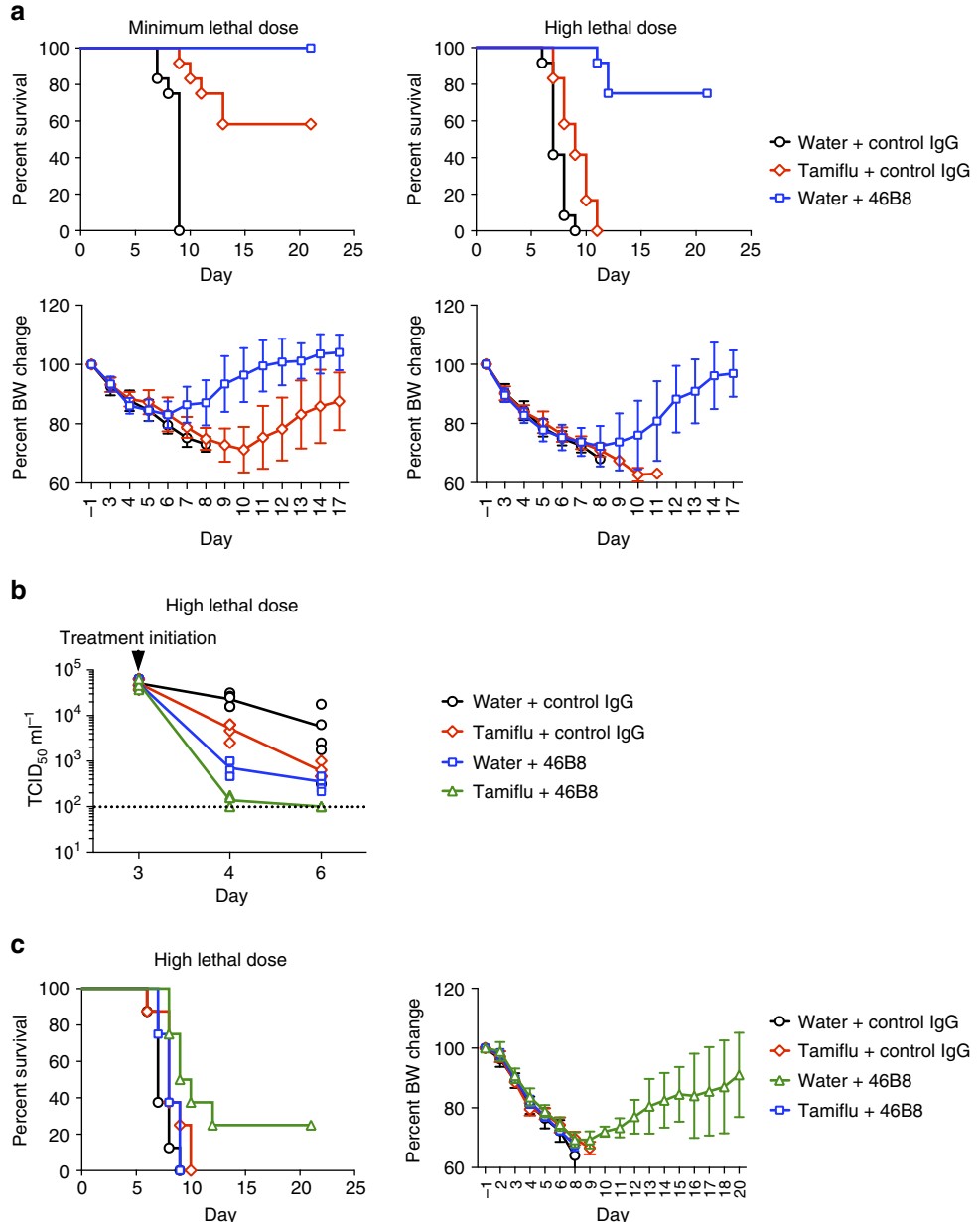

**Figure 5 | *In vivo* efficacy of 46B8 compared with and in combination with Tamiflu.** (**a**) DBA/2 J mice were infected intranasally with a minimum lethal dose (left panels) or a high lethal dose (right panels) of B/Victoria/504/2000. At 72 h post infection, mice received a single treatment of 46B8 or a control IgG intravenously at 45 mg kg$^{-1}$, or Tamiflu orally at 100 mg kg$^{-1}$ twice a day for 5 days. Top panels: survival curves. Each group contains 12 mice. Log-rank tests of 46B8- versus Tamiflu-treated groups in both infection models give $P < 0.05$. Bottom panels: percent of average BW of survived mice as compared with the average pre-infection weight. (**b**) DBA/2 J mice were infected intranasally with a high lethal dose of B/Victoria/504/2000. At 72 h post infection, mice received a single treatment of 46B8 or a control IgG intravenously at 45 mg kg$^{-1}$, Tamiflu orally at 100 mg kg$^{-1}$ twice a day for 3 days, or a combined treatment of 46B8 and Tamiflu. At day 3, 4 and 6 post infection, lung homogenates were prepared and viral titres in the homogenates were determined with a TCID$_{50}$ assay on MDCK cells. Titres are shown in TCID$_{50}$ ml$^{-1}$. Each group contains 10 mice. An unpaired two-tailed *t*-test of 46B8 versus Tamiflu and the combined treatment versus 46B8 or Tamiflu alone at day 4 and 6 give $P < 0.05$. Dotted line: detection limit of the assay; arrow: treatment initiation point. (**c**) DBA/2 J mice were infected intranasally with a high lethal dose of B/Victoria/504/2000. At 72 h post infection, mice received a single treatment of 46B8 or a control IgG intravenously at 5 mg kg$^{-1}$, Tamiflu orally at 100 mg kg$^{-1}$ twice a day for 5 days, or a combined treatment of 46B8 and Tamiflu. Left panel: survival curves. Each group contains eight mice. A log-rank test of the combined treatment versus 46B8 alone gives $P < 0.05$, and the combined treatment versus Tamiflu alone gives $P = 0.06$. Right panel: percent of average BW of survived mice as compared with the average pre-infection weight. (**a**,**c**: mean and s.d.)

B/Victoria/504/2000 mouse model. Infected mice were treated at 72 h post infection with either 46B8 (45 mg kg$^{-1}$), Tamiflu (100 mg kg$^{-1}$) twice a day for 3 consecutive days, or a combination of these two therapies. Because 46B8 treatment alone is highly efficacious leaving little room for observing

improvement with the survival curve, we measured viral titres in the lung early after treatment. All groups showed similar lung titres at day 3 post infection when treatment started (Fig. 5b). As expected, 46B8 alone significantly reduced lung titres at day 4 and 6 compared with Tamiflu alone. Combined treatment further

reduced the titres significantly, almost to the detection limit of the assay. In order to see any additive effect of 46B8 and Tamiflu in mouse survival, we tested a sub-efficacious dose of 46B8 (5 mg kg$^{-1}$). Infected mice were treated at 72 h post infection with either 48B8 (5 mg kg$^{-1}$), Tamiflu (100 mg kg$^{-1}$) twice a day for 5 consecutive days, or a combination of the two therapies (Fig. 5c). Mice receiving either 46B8 or Tamiflu alone exhibited 100% mortality by day 10, similarly to the control group. However, co-administration of 46B8 with Tamiflu improved outcome over either treatment alone, resulting in 25% survival (Fig. 5c, left panel). Consistent with the survival benefit, the surviving mice were able to regain 90% of their pre-infection BW before the end of the study (Fig. 5c, right panel). These results suggest that co-administration of 46B8 and Tamiflu could improve the therapeutic benefits of either treatment alone.

**Selection and characterization of 46B8-resistant viruses.** To select viruses that are resistant to 46B8, we passaged B/Brisbane/ 60/2008 in the presence of increasing concentrations of 46B8 on Madin-Darby Canine Kidney Epithelial (MDCK) cells. After eight rounds of passage, three independent resistant clones (A4, B1 and C2) emerged. All three resistant viruses escaped neutralisation by 46B8 in a plaque reduction assay (Fig. 6a). Interestingly, A4 appeared to be more resistant to 46B8 or 46B8 N297G than B1 and C2 in a neutralisation assay that detects the expression of viral nucleoprotein at 16 h post infection (Supplementary Fig. 10). Sequence analysis revealed a common mutation, Ser301Phe (S301F, numbering includes the signal sequence), in the HA proteins of all three viruses. B1 and C2 contain an additional mutation in HA, Lys401Glu (K401E). Ser301 in B/Brisbane/60/ 2008 HA corresponds to Ser300 in B/Victoria/504/2000 HA, which is a key epitope residue at the centre of the 46B8-binding site (Fig. 3b). Lys401 (Lys400 in B/Victoria/504/2000 HA) is in HA stalk, outside the 46B8 contact region (Supplementary Fig. 8).

To study the effect of the mutations on 46B8 binding, we expressed single (S301F or K401E) or double (S301F/K401E)

mutant HAs on cell surface and tested 46B8 binding by flow cytometry. As shown in Fig. 6b, both WT and mutant HAs bound 34B5 similarly, indicating comparable expression levels (left panel). The K401E mutant bound 46B8 as well as the WT HA, whereas S301F and S301F/K401E showed slightly reduced binding at pH 7.0 (middle panel). At pH 4.8, both WT and K401E HAs bound 46B8 significantly, whereas S301F and the double mutant completely lost binding (Fig. 6b, right panel). The almost identical binding profiles between WT and K401E and between S301F and S301F/K401E suggest that K401E is likely an inconsequential random mutation. These binding results also suggest that the mutant viruses escaped 46B8 via abolishing antibody binding at low pH in endosomes/lysosomes. 46B8, however, retained the ability to bind mutant HAs at pH 7.0, as evidenced by immunogold-EM studies: 34B5, 46B8 and 46B8 N297G all bound efficiently to both the WT and mutant viruses, whereas a control IgG did not bind (Supplementary Fig. 11).

We next examined the fitness of the 46B8-resistant viruses *in vitro* and *in vivo*. First, we infected MDCK cells with WT or mutant viruses at a low multiplicity of infection (MOI) of 0.01 and measured the amount of released viral genome in the supernatant by qPCR. At day 4 post infection, all three mutant viruses yielded slightly more progeny viruses than WT, not showing any replication defect (Fig. 7a). Second, we infected mice with these viruses at different doses and monitored mouse survival and BW. All three mutant viruses caused 100% mortality at rates comparable to WT, indicating no reduced fitness *in vivo* (Fig. 7b, top panels). Consistent with the survival results, mice infected by either the WT or mutant viruses lost weight to similar degrees before they died (Fig. 7b, bottom panels). These results suggest that despite the highly conserved nature of S301 (99.3%), viruses carrying the F301 allele do not exhibit reduced fitness *in vitro* or in mice.

Although the 46B8-resistant viruses possess WT-like fitness, they should be sensitive to Tamiflu because 46B8 and Tamiflu target distinct viral proteins and different steps of the viral life cycle, entry and release, respectively. To test this hypothesis, we

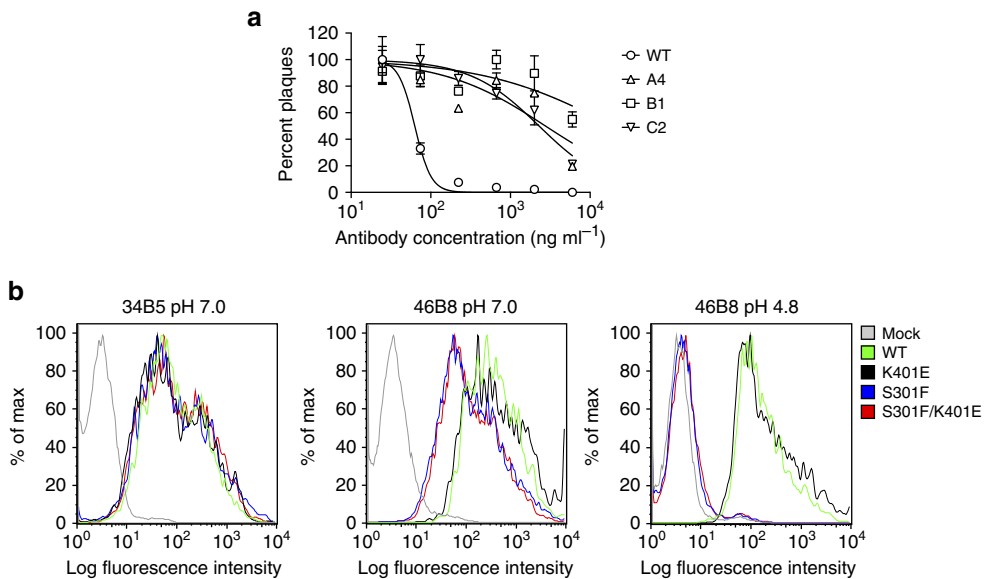

**Figure 6 | Properties of mutant B/Brisbane/60/2008 viruses and HAs.** (**a**) Plaque reduction assay. 100 plaque-forming units (pfu) of each virus was incubated with varying concentrations of 46B8 for 1 h prior to infection of MDCK cells. At 1 h post infection, the inoculum was removed and cells were overlaid with varying concentrations of 46B8 in agarose. At day 4 post infection, the numbers of plaques were counted for each virus and normalized to the number at the lowest antibody concentration. The assay was done in triplicate. (**b**) 293 T cells expressing the WT or mutant B/Brisbane/60/2008 HAs were incubated with 34B5 (left panel) or 46B8 (middle and right panels) at pH 7.0 (left and middle panels) or 4.8 (right panel). Flow cytometry profiles are shown. Mock, mock transfected cells. (**a**: mean and s.e.m.)

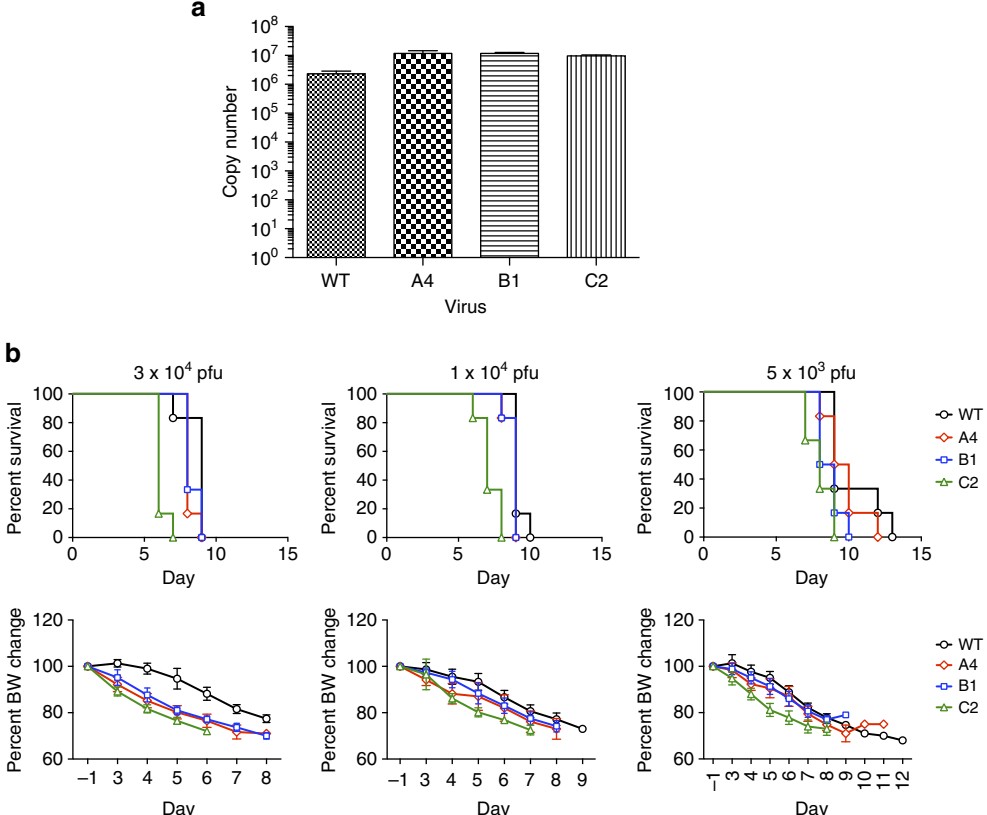

**Figure 7 | Fitness of mutant B/Brisbane/60/2008 viruses.** (**a**) *In vitro* fitness. MDCK cells were infected with the WT or mutant B/Brisbane/60/2008 viruses at MOI 0.01. Released viral genome in the supernatant was quantitated at day 4 post infection by qPCR of the viral M1 Matrix gene. Genome copy numbers are shown as histograms. The assay was done in triplicate. (**b**) *In vivo* fitness. DBA/2 J mice were infected with WT or mutant viruses at $3 \times 10^4$, $1 \times 10^4$ or $5 \times 10^3$ plaque-forming units (pfu) per mouse. Survival curves (top panels) and percent of average BW as compared with the average pre-infection weight (bottom panels) are shown. Each group contains six mice. (**a**: mean and s.e.m.; **b**: mean and s.d.)

infected MDCK cells with the viruses and added oseltamivir acid (the active form of Tamiflu) during the culture prior to measuring released viral genome by qPCR. As shown in Fig. 8a, oseltamivir acid inhibited the mutant viruses in a dose-dependent manner similarly to the WT virus. Furthermore, when mice were infected with the minimum lethal doses of these viruses and treated with Tamiflu (100 mg kg$^{-1}$ twice a day for 5 days) starting at 48 h post infection, Tamiflu fully protected mice infected with the WT or mutant viruses with the exception of a single mouse in the B1 group (Fig. 8b). All groups of mice were able to regain 100% of their pre-infection BW (Supplementary Fig. 12a).

Next, we tested whether the mutant viruses were resistant to 46B8 *in vivo*. Mice were infected with the minimum lethal dose of the WT or mutant viruses and treated with 46B8 (15 mg kg$^{-1}$) at 72 h post infection. As expected, a single 46B8 treatment was able to protect 100% of mice infected with the WT virus (Fig. 8c). Importantly, with the exception of a single mouse in the C2 group, all mice infected with the mutant viruses were also fully protected by 46B8. Consistent with the survival benefits, all groups of mice that survived infection were able to regain 100% of their pre-infection BW (Supplementary Fig. 12b). Mice infected with the mutant viruses however showed more BW loss and recovered weight more slowly than those infected with the WT virus.

As the mutant viruses are resistant to 46B8 *in vitro*, we wondered how they could be sensitive to 46B8 *in vivo*. Given that 46B6 is capable of eliciting ADCC and retained the ability to bind mutant HAs on cell surface at pH 7.0 (Fig. 6b), we hypothesized that it could protect mice against the mutant viruses via its Fc

functions. To test this hypothesis, we first performed the *in vitro* ADCC assays on target cells infected with the WT or mutant viruses and then coated with 46B8. All target cells induced CD107a expression on NK cells to similar levels and were lysed by activated NK cells to similar degrees (Fig. 9a), suggesting that 46B8 is capable of inducing ADCC and lysing cells infected with the mutant viruses.

Next, we tested whether the Fc functions are required for *in vivo* protection. Mice were infected with the minimum lethal dose of the WT or mutant viruses and treated with 15 mg kg$^{-1}$ of 46B8, 46B8 N297G or a control IgG at 72 h post infection. As expected, mice infected with the WT or mutant viruses were protected by 46B8 but not the control IgG (Fig. 9b). 46B8 N297G fully protected mice against the WT virus but completely lost efficacy against the A4 mutant, supporting an important role of Fc functions in *in vivo* protection. Interestingly, 60% of the mice infected with the B1 or C2 mutant were protected by 46B8 N297G, possibly due to the partial sensitivity of these two viruses to the mAb (Supplementary Fig. 10). Consistent with the survival benefits, all groups of mice that survived infection were able to regain 100% of their pre-infection BW (Fig. 9c).

As little difference in mAb binding was detected between the mutant HAs (Fig. 6b) or viruses (Supplementary Fig. 11), we performed whole-genome sequencing of the viruses to identify mutations in viral proteins other than HA that might contribute to the differential sensitivities of the mutant viruses to 46B8. As summarized in Supplementary Table 1, three most dominant ($>99\%$) mutations unique to A4 (R185K) or B1 and C2 (I303M and E582G) were identified in the PB2 protein.

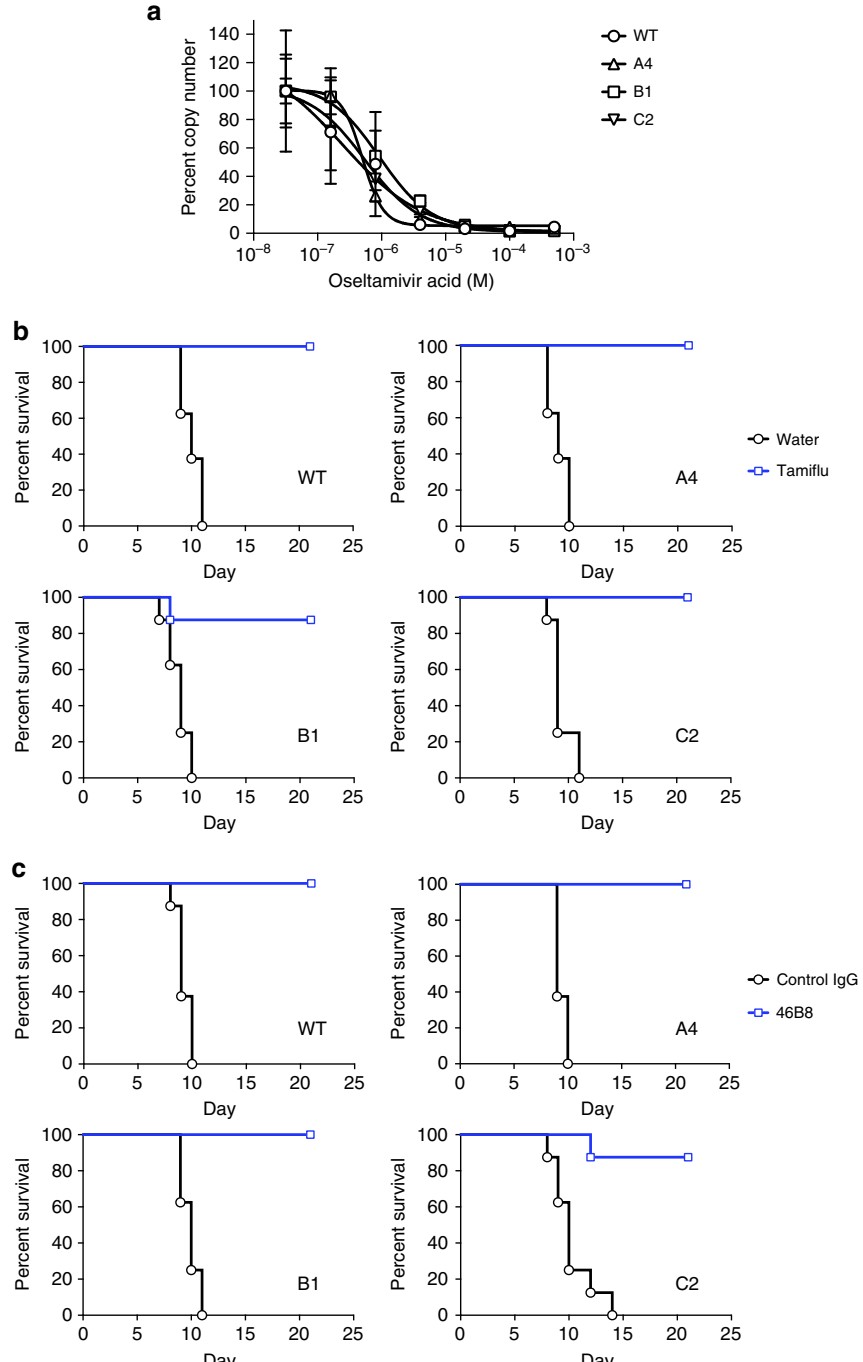

**Figure 8 | Both WT and mutant B/Brisbane/60/2008 viruses are sensitive to 46B8 *in vivo* and to Tamiflu.** (**a**) MDCK cells were infected with WT or mutant viruses at MOI 0.01. After removal of the virus inoculum, cells were grown in the presence of varying concentrations of oseltamivir acid for 4 days. Released viral genome in the supernatant was quantitated by qPCR of the viral M1 Matrix gene. The copy numbers of each virus were normalized to the value at the lowest oseltamivir acid concentration. The assay was done in triplicate. (**b**) DBA/2 J mice were infected intranasally with a minimum lethal dose of the WT or mutant viruses. At 48 h post infection, mice received Tamiflu orally at 100 mg kg$^{-1}$ twice a day for 5 days. Survival curves are shown. Each group contains eight mice. Log-rank tests of Tamiflu versus water treated groups for all viruses give $P<0.05$. (**c**) DBA/2 J mice were infected intranasally with a minimum lethal dose of the WT or mutant B/Brisbane/60/2008 viruses. At 72 h post infection, mice received a single treatment of 46B8 or a control IgG intravenously at 15 mg kg$^{-1}$. Survival curves are shown. Each group contains eight mice. Log-rank tests of 46B8 versus control IgG-treated groups for all viruses give $P<0.05$. (**a**: mean and s.e.m.)

These substitutions were not reported previously and are not likely to impair viral replication or virulence in mouse, as the mutant viruses did not show reduced viral fitness *in vitro* or *in vivo* (Fig. 7). Two other highly dominant mutations (K338R and V625A) were identified in the PA protein. As they are not unique to A4 or B1 and C2, they are also not likely to contribute to the differential sensitivities to 46B8. Other less-dominant mutations were identified in the PB1, PB2, PA, neuraminidase, BM2 and NS1 proteins. Their effects on viral life cycle are not known.

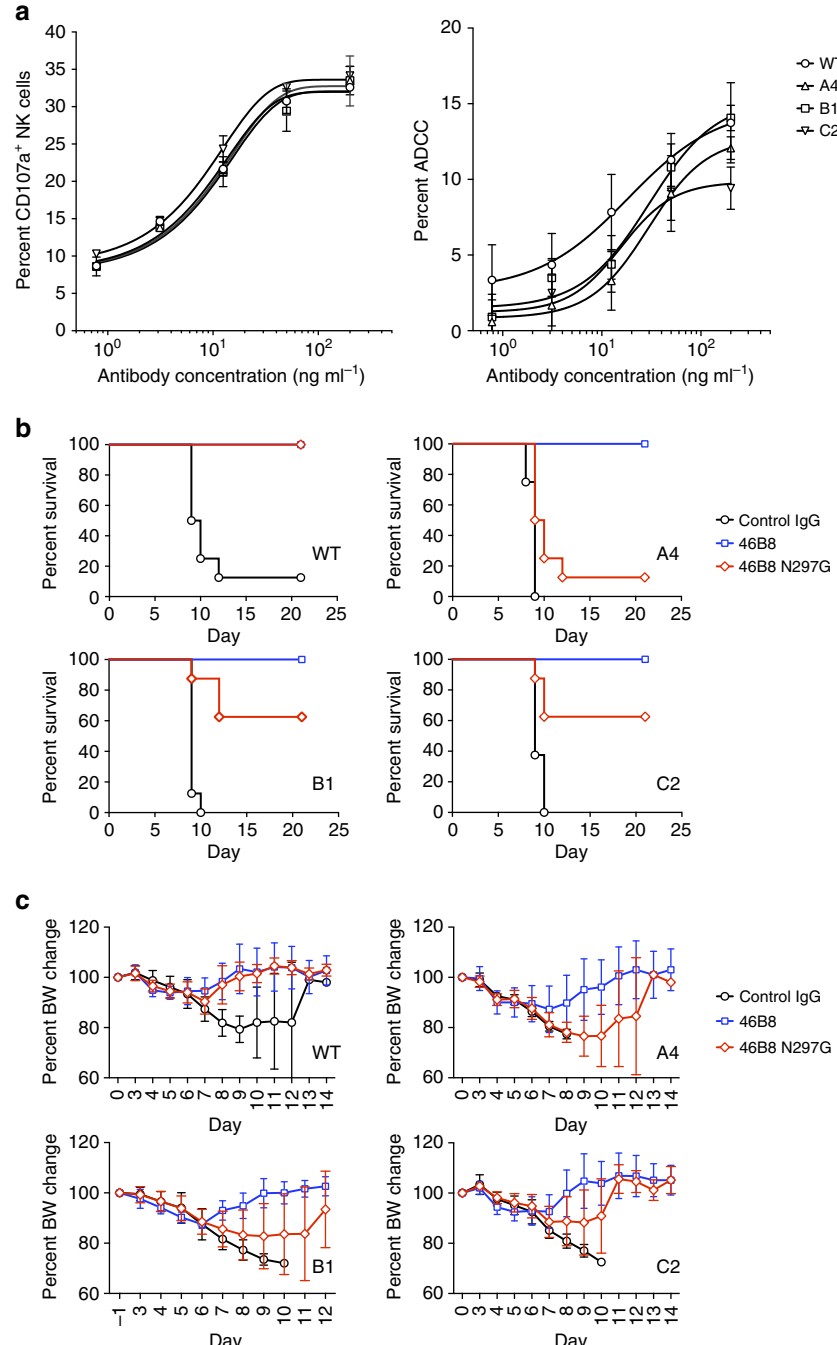

**Figure 9 | 46B8 induces ADCC via both WT and mutant B/Brisbane/60/2008 HAs. (a)** A549 cells infected with WT or mutant B/Brisbane/60/2008 viruses were labelled with 46B8 prior to incubation with NK cells. NK cell activation (left panel) and target cell lysis (right panel) were presented as the frequency of CD107a[+] cells and the percent of LDH release, respectively. The assays were done in duplicate. Results are representative of three independent experiments. **(b,c)** DBA/2 J mice were infected intranasally with a minimum lethal dose of the WT or mutant B/Brisbane/60/2008 viruses. At 72 h post infection, mice received a single treatment of 46B8, 46B8 N297G or a control IgG intravenously at 15 mg kg$^{-1}$. **(b)** Survival curves. Each group contains eight mice. Log-rank tests of 46B8- versus control IgG-treated groups for all viruses, 46B8 N297G- versus control IgG-treated groups for WT, B1 and C2 and 46B8- versus 46B8 N297G-treated groups for A4, B1 and C2 give $P < 0.05$. **(c)** Percent of average BW of survived mice as compared with the average pre-infection weight. (**a**: mean and s.e.m.; **c**: mean and s.d.)

## Discussion

We identified a rare human anti-IBV mAb, 46B8, that neutralized all IBV strains tested *in vitro*. Importantly, it fully protected mice from lethal challenges of IBVs from both lineages and the ancestral strains, even when given at 72 h post infection. Consistent with its prolonged *in vivo* efficacy window, a single treatment of 46B8 demonstrated superior protection over a 5-day

Tamiflu regimen in mice challenged with either a minimum or a high lethal dose of IBV. Interestingly, co-administration of 46B8 and Tamiflu improved the therapeutic outcome of either treatment alone, likely because they target distinct viral proteins and different steps of the viral life cycle. Because of its superior therapeutic efficacy over Tamiflu, the current standard of flu treatment, 46B8 is a promising candidate for the development of

new anti-IBV medicines, especially as a late treatment option or a combination therapy with Tamiflu.

46B8 recognizes a highly conserved epitope in the vestigial esterase domain. It blocks low pH-induced conformational changes in HA during membrane fusion. Structural analysis revealed two factors that could contribute to the neutralisation mechanism and efficacy of 46B8. First, 46B8 binds to HA at a slant angle tilting upward. This binding angle might increase the accessibility of 46B8 to HA on the highly packed viral surface[26,27], resulting in its high protective efficacy in mice. Second, EM density from the top view showed that 46B8 spans a cleft formed by two adjacent HA monomers and possibly cross-link them. Consequently, 46B8 was able to prevent the dissociation of the HA head domains preceding the conformational changes in the stalk region during low pH-induced membrane fusion[28–30]. This mechanism is similar to that of F005-126, a broadly neutralising mAb against H3N2 IAVs that binds to the esterase domain and blocks low pH-induced conformational changes in HA[31]. By contrast, another esterase-binding and neutralising mAb against IAV, HC45, binds only one HA monomer and does not block HA conformational changes[32]. Instead, HC45 inhibits viral attachment to cell-surface receptors despite binding to the esterase domain, possibly because it tilts sharply upward toward the RBP and imposes steric hindrance to receptor access given the large size of IgG[33]. By contrast, 46B8 also tilts upward but at a smaller angle that positions it farther away from the RBP, which might account for the lack of HI activity of 46B8. Interestingly, HC63, an anti-IAV mAb targeting the RBP and inhibiting viral attachment to receptors[32], is also able to block low pH-induced HA conformational changes because it binds simultaneously to the globular heads of two adjacent HA monomers and cross-links them[34]. Despite its importance in preventing the dissociation of HA1 heads, cross-linking of two HA monomers might not be absolutely required for esterase domain-binding mAbs to block low pH-induced HA conformational changes. H5M9, a broadly neutralising mAb against H5N1 IAVs, appears to bind to the esterase domain of only one HA monomer but is still able to block HA conformational changes[35]. This mAb also demonstrated HI activity, consistent with its sharp upward binding angle[35,36]. Together, these findings provided insights to the major neutralising mechanisms used by esterase domain-binding mAbs: (a) interfere with receptor binding by tilting upward close to the RBP, (b) block membrane fusion either by cross-linking two HA monomers, and thus preventing HA1 dissociation, or by blocking HA conformational changes using a mechanism that has yet to be fully elucidated and (c) inhibit viral egress.

We isolated three 46B8-resistant viruses that all harbour a S301F mutation. Residue S301 is in the middle of the 46B8-binding site, supporting the EM results. The 46B8-resistant viruses do not show reduced viral fitness in vitro or in vivo, possibly because the function of the ancestral esterase domain is replaced by the activity of the neuraminidase protein[37], and mutations in the vestigial esterase domain are tolerable to the virus. By contrast, resistant viruses to stalk-binding anti-IAV mAbs showed reduced viral fitness, likely due to the critical functions of the stalk domain in low pH-induced membrane fusion during viral entry[38–40].

All three 46B8-resistant viruses were equally sensitive to Tamiflu, consistent with the distinct viral targets of 46B8 and Tamiflu. Interestingly, 46B8 was able to protect mice against lethal challenges of the mutant viruses despite their resistance in vitro, indicating the contribution of Fc effector functions in protection in vivo. Mice infected with the mutant viruses however showed more BW loss and recovered weight more slowly than those infected with the WT virus, likely because the mutant

viruses can still enter cells and achieve productive infection in the presence of 46B8, which resulted in elevated immune responses and more severe BW loss. In support of a role of Fc functions in mouse protection by 46B8, we demonstrated that 46B8 is able to induce ADCC and kill both WT and mutant virus-infected cells in vitro. Furthermore, 46B8 N297G, the Fc-defective mAb that was not able to induce ADCC in vitro, also lost efficacy against the mutant viruses in mouse, indicating the critical role of Fc functions in in vivo protection when an mAb loses its ability to neutralize virus via its Fab domain. In addition to ADCC, other Fc-dependent effector functions such as antibody-dependent cellular phagocytosis[41,42], antibody-dependent respiratory burst[42] and antibody-dependent NETosis[43,44] could also contribute to the in vivo protection by 46B8. Given the highly conserved nature of IBV esterase domain, broadly neutralising mAbs targeting this domain are promising candidates for new anti-IBV therapies. Compared with mAbs targeting the RBP of HA, esterase domain-binding mAbs such as CR8071 do not interfere with HA binding to sialic acid receptors on effector cells and are more likely to induce Fc-dependent effector functions[15]. However, owing to the less-functional importance of the esterase domain compared with the HA stalk, antibody-resistant viruses might arise. Our findings underline the importance of having both Fab- and Fc-dependent antiviral mechanisms, especially for esterase domain-binding mAbs to combat potential resistant viruses.

## Methods

**Antibody discovery.** Plasmablast isolation from vaccinated human donors, in vivo expansion and enrichment of antigen-reactive plasmablasts, antigen-specific cell sorting by flow cytometry and IgG cloning from single plasmablasts were described previously[16,17]. In brief, peripheral blood mononuclear cells were isolated from human blood or leukopac and mixed with IBV HA protein to activate antigen-specific cells prior to intrasplenic transplantation into 6–8 weeks old female SCID mice (Charles River Laboratories) for rapid expansion and enrichment of human plasmablasts. Individual IBV HA-specific plasmablasts were isolated from splenic cells of the SCID mice by FACS and subjected to IgG cloning by reverse transcription and PCR[16]. The variable regions of the heavy and light chains were amplified and inserted into the pRK (pF8CIS9080) mammalian expression vectors[45] containing standard human IgG1 heavy and light chain constant regions. Plasmids encoding heavy and light chains were co-transfected into human embryonic kidney epithelial 293 T cells (ATCC, CRL-3216) and secreted mAbs were subjected to ELISA screening. The Fluvirin seasonal influenza vaccine (Novartis) contained the B/Brisbane/60/2008 strain of IBV. Recombinant soluble HA protein from the B/Hong Kong/8/1973 strain was generated with synthesized coding DNA in the pRK vector and used for enrichment and sorting of IBV HA-specific plasmablasts. Plasmids encoding the B/Maryland/1/1959, B/Victoria/504/2000 or B/Brisbane/60/2008 HA proteins were generated with synthesized coding DNA in the pRK vector. Lysates of 293 T cells transfected with these plasmids were used for ELISA screening of cloned IgGs. Leukopac or blood from healthy human donors was obtained after written informed consent was provided and ethical approval granted from the Western Institutional Review Board.

**Neutralisation assay.** MDCK cells (ATCC, CCL-34) were grown as a 50% confluent monolayer in 96-well black imaging plates with clear bottom. IBVs were purchased from ViraPur (San Diego) and diluted in influenza media (DMEM + 0.2% BSA + 2 µg ml$^{-1}$ TPCK-treated trypsin) to an MOI of 1 and incubated for 1 h at 37 °C with 50 µg ml$^{-1}$ of different mAbs. The mAb/virus cocktail was allowed to infect the MDCK cells for 16 h at 37 °C in a 5% $CO_2$ incubator prior to fixation with cold 100% ethanol. The fixed cells were stained with Hoechst 33342 (Life Technologies) to visualize cell nuclei and with a mAb specific for IBV nucleoprotein (Millipore) followed by an Alexa488-conjugated secondary antibody to detect virus-infected cells. Cells were imaged using an Image Express Micro apparatus (Molecular Devices) and images were generated and analysed with the MetaXpress 3.1 software. The percentages of infected cells (normalized to the lowest antibody concentration) were determined and plotted on the y axis versus the Log10 antibody concentrations on the x axis. The data were fit with a nonlinear regression dose–response curve using the GraphPad Prism v.6.0 software.

**ELISA.** To generate full-length HA lysates or His-tagged soluble HA (ectodomain) proteins, 293 T cells were transfected with HA-expressing plasmids. Forty-eight hours later, supernatants were collected and soluble proteins were purified with a Ni column. Cells were treated with lysis buffer (50 mM Tris pH 8, 5 mM EDTA pH

8, 150 mM NaCl, 1% Triton X-100, protease inhibitor cocktail) for 20 min at room temperature and lysates were centrifuged at 14,000 r.p.m. (20,000g) for 10 min. Supernatants were stored at $-80\,^{\circ}\mathrm{C}$ and used in ELISA. For ELISA, Maxisorp 96-well plates (Nunc) were coated with $5\,\mu g\,ml^{-1}$ of *Galanthus nivalis* lectin (Sigma) in PBS for 6 h at room temperature. Plates were washed with Washing Buffer (PBS pH 7.4 + 0.05% Tween-20) and incubated in Blocking Buffer (PBS pH 7.4 + 0.5% BSA) for 1 h at room temperature. Plates were washed and incubated with HA lysates (1:300) or proteins ($10\,\mu g\,ml^{-1}$) in Assay Diluent (PBS pH 7.4 + 0.5% BSA + 0.05% Tween-20) overnight at $4\,^{\circ}\mathrm{C}$. Plates were washed and incubated with serially diluted mAbs (threefold dilutions ranging from 3,000 to $1.37\,ng\,ml^{-1}$) in Assay Diluent for 1.5 h at room temperature. Plates were washed and incubated with a goat-anti-human-IgG-HRP secondary antibody (Jackson ImmunoResearch) at 1:30,000 in Assay Diluent for 1 h at room temperature. Plates were washed and incubated with the TMB Substrate (KPL) for 5–10 min at room temperature prior to the addition of 1M phosphoric acid to stop the reaction. Absorbance at 450 nM was measured on a BioTek Synergy2 plate reader with the Gen5 software.

**Plaque reduction assay.** MDCK cells were grown in six-well tissue culture plates and infected with serial dilutions of IBVs in influenza media (DMEM + 0.2% BSA + $2\,\mu g\,ml^{-1}$ TPCK-treated trypsin). One hour after infection, viruses were removed and cells were overlaid with influenza media + 1% agarose. Infected cells were incubated at $37\,^{\circ}\mathrm{C}$ for 4–6 days and the numbers of plaques were counted by eye or under microscope. For each virus, the amount of viral stock that gives ~100 plaques (100 plaque-forming units or 100 pfu) was used for plaque reduction assay. In brief, virus was incubated with varying concentrations of 46B8 in influenza media for 1 h prior to infection of cells. At 1 h post infection, the inoculum was removed and cells were overlaid with varying concentrations of 46B8 in influenza media + 1% agarose and incubated at $37\,^{\circ}\mathrm{C}$ for 4–6 days until plaques were visible. The numbers of plaques were counted by eye or under microscope. For each virus, the percent of plaques (normalized to the number of plaques at the lowest antibody concentration) was plotted on the y axis versus the Log10 antibody concentrations on the x axis. The data were fit with a nonlinear regression dose–response curve with the Prism 6.0 software (GraphPad) to generate the IC50 and 95% CI data. The assay was done in triplicate with data presented as mean ± s.e.m.

**HI assay.** Hemagglutination assay was carried out on turkey red blood cells (TRBCs, Lampire Biological Laboratories). For each virus, eight serial dilutions in fivefold steps were made in triplicates in PBS, starting at 1:5. 50 μl of each dilution was transferred to V-bottom 96-well plate. TRBCs were diluted to 0.5% in PBS and 50 μl was added to each well containing virus. The plate was incubated at room temperature for 3 h. The last virus dilution (or lowest virus concentration) that prevented TRBC precipitation was determined by direct visualisation and used for HI assay.

For HI assay, serial dilutions of each antibody in triplicate were mixed with pre-determined amount (above) of B/Victoria/504/2000 or B/Wisconsin/1/2010 virus in PBS and incubated at $37\,^{\circ}\mathrm{C}$ for 1 h. In total, 50 μl of the virus-antibody mixture was transferred to V-bottom 96-well plate. TRBCs were diluted to 0.5% in PBS and 50 μl was added to each well containing the virus-antibody mixture. The plate was incubated at room temperature for 3 h and HI titre (the lowest antibody concentration that inhibited viral agglutination and allowed TRBC precipitation) was determined for each antibody by direct visualisation.

**Neutralisation of pseudotype virus.** HIV pseudotype virus bearing the B/Wisconsin/1/2010 HA I196 or T196 surface protein was produced as previously described[16]. In brief, 293 T cells were co-transfected with three plasmids: pCMV-dR8.9 (ref. 46), pHR'CMVGFP[47] and an HA-expressing plasmid generated with synthesized coding DNA in the pRK vector. Two days later, culture supernatant was collected and filtered through a 0.45-μm filter. Pseudotype viruses in the supernatant were purified by ultra-centrifugation through 20% sucrose. For neutralisation assay, pseudotype viruses were incubated with serial dilutions of 34B5 for 1 h before adding to target 293 T cells in collagen-coated 96-well plates. Forty-eight hours later, the numbers of infected cells were determined by counting green fluorescent protein-positive cells. Infection of each virus was normalized to the number of infected cells at the lowest antibody concentration. Percent of infection was plotted against the antibody concentrations. Data were fit with a nonlinear regression dose–response curve. The assay was done in duplicate with data shown as mean ± s.e.m.

**HA0 activation inhibition assay.** Recombinant soluble B/Hong Kong/8/1973 HA protein (0.6 μg) was incubated with PBS, a control IgG against the glycoprotein D of human cytomegalovirus[16], 34B5 or 46B8 ($1\,mg\,ml^{-1}$) for 45 min at $37\,^{\circ}\mathrm{C}$ prior to exposure to TPCK-treated trypsin ($2.5\,\mu g\,ml^{-1}$) for 2 or 10 min at $37\,^{\circ}\mathrm{C}$. Trypsin digestion was stopped by addition of the LDS sample buffer (Life Technologies). Samples were run on SDS-PAGE gel (reduced) and blotted according to standard methods. HA bands were detected by a mouse mAb that recognizes the HA2 subunit of IBV HA (Novus Biologicals).

**Fusion inhibition assay.** Hela cells (ATCC, CCL-2) were grown in DMEM + 10% FBS to ~40% confluent in six-well tissue culture plates and transfected with 5 μg of plasmid encoding the B/Wisconsin/1/2010 HA. Two days later, cells were washed with PBS and treated with $5\,\mu g\,ml^{-1}$ of TPCK-treated trypsin in PBS for 5 min at $37\,^{\circ}\mathrm{C}$. Trypsin was removed and cells were incubated with culture media containing $25\,\mu g\,ml^{-1}$ of soybean trypsin inhibitor (CalBiochem) for 10 min at $37\,^{\circ}\mathrm{C}$. Cells were then incubated with $200\,\mu g\,ml^{-1}$ of the control IgG, 34B5 or 46B8 for 1 h at $37\,^{\circ}\mathrm{C}$ prior to exposure to influenza media adjusted to pH 4.8 for 2 min at $37\,^{\circ}\mathrm{C}$ to induce fusion. Cells were washed with DMEM and grow in culture media overnight to allow full formation of syncytia. Phase images of the cells were acquired under the ×10 objective on a Nikon Eclipse TE2000-E microscope with the NIS-Elements AR3.2 software.

**HA conformational change assay.** 293 T cells were transfected with plasmid encoding the B/Victoria/504/2000 HA. Two days later, cells were collected in a PBS-based enzyme-free cell dissociation buffer (Life Technologies) and washed with PBS. Cells were treated with $2.5\,\mu g\,ml^{-1}$ of TPCK-treated trypsin in PBS for 2 min and neutralized with $25\,\mu g\,ml^{-1}$ of soybean trypsin inhibitor. Cells were then incubated with 46B8 or the control IgG for 1 h prior to exposure to a pH 4.8 buffer for 2 min at $37\,^{\circ}\mathrm{C}$ to induce HA conformational change. Cells were washed and incubated with 50 mM DTT or PBS for 20 min to allow the dissociation of the HA1 subunit. Cells in all treatment groups were tested for 34B5 binding by flow cytometry and results were analysed with the FlowJo 8.4.5 software. The mean fluorescence intensities were normalized to the PBS-treated cells in each group and the percent of binding was plotted as histograms with the Prism 6.0 software.

***In vitro* ADCC assay and CDC assay.** A549 lung epithelial cells (ATCC, CCL-185) were infected with WT or mutant B/Brisbane/60/2008 viruses at a MOI of 10 for 18 h and used as target cells. For ADCC assay, NK cells were isolated from fresh human blood by negative selection with the RosetteSep Human NK Cell Enrichment Cocktail (STEMCELL Technologies) and used as effector cells. Target cells were incubated with varying concentrations of 46B8, 46B8 N297G (an Fc mutant that does not engage FcγR generated by site-directed mutagenesis of 46B8), Rituximab[20] (a negative control mAb that binds to CD20) or Cetuximab[21] (a positive control mAb that binds to EGFR) for 30 min. For NK cell activation assay, effector cells were mixed with antibody-bound target cells at a 1:1 ratio for 3 h prior to flow cytometry analysis of surface CD107a (LAMP-1) expression, a marker of degranulation. The percent of CD107a-positive cells (as compared with control unstimulated NK cells) was plotted against antibody concentrations and data were fit with a nonlinear regression dose–response curve with the Prism 6.0 software. For target cell lysis ADCC assay, effector cells were mixed with antibody-bound target cells at a 5:1 ratio for 4 h. Culture supernatants were mixed with the LDH substrate in the Cytotoxicity Detection Kit (Roche). Absorbance at 490 nm was measured on a SpectraMax plate reader (Molecular Devices). Signal of wells containing only the antibody-bound target cells represented spontaneous release of LDH (spontaneous release); wells containing target cells lysed by Triton X-100 provided the maximum signal available (maximum lysis). Antibody-independent cellular cytotoxicity (AICC) was measured in wells containing the effector cells and target cells without bound antibody. The extent of specific ADCC was calculated using the following equation: %ADCC = 100 × ((sample signal-AICC signal)/ (maximum lysis signal-spontaneous release signal)). The percent of ADCC was plotted against antibody concentrations and data were fit with a nonlinear regression dose–response curve with the Prism 6.0 software.

For CDC assay, virus-infected A549 cells or WIL2-S cells (ATCC, CRL-8885, expressing CD20) were coated with varying concentrations of 46B8 or Rituximab. Human serum complement (Quidel, A-400) was added to antibody-coated cells and incubated for 2 h at 37C to allow cell lysis. CellTiter-Glo reagent (Promega) was then added to detect ATP in the remaining live cells that were not lysed by CDC. Luminescence signals were measured as random luminescence unit using a SpectraMax plate reader and plotted against antibody concentrations. Data were fit with a nonlinear regression dose–response curve with the Prism 6.0 software.

**EM sample preparation and image processing.** For expression of the 46B8 Fab fragment, constructs containing the 46B8 heavy chain (VH/CH1) and light chain (VL/CL) were generated in the pRK vector. For expression of soluble HA, cDNA corresponding to residues 1–530 of B/Victoria/504/2000 HA (GenBank ID AAT69425.1) ectodomain was cloned into the same expression vector with the native signal peptide and an in-frame C-terminal ×6 His tag. 46B8 Fab, HA and 46B8/HA complex were expressed in 293 S cell, a cell line adapted for growth in suspension from the parental 293 cell (ATCC, CRL-1573), by transient transfection and collected after 7 days. The 46B8 Fab fragment alone was purified on a Protein G sepharose column, washed with Tris-buffered saline (TBS, 20 mM TrisHCl pH 8, 150 mM NaCl), eluted with 0.1M glycine pH 3.0 and neutralized with 0.2 M TrisHCl pH 8.0. HA and the 46B8/HA complex were purified on Ni Sepharose FF, washed with TBS + 20 mM imidazole, and eluted with TBS + 200 mM imidazole. The 46B8/HA complex was concentrated and further purified by size exclusion chromatography on a S200 10/30 column in TBS. Fractions that corresponded to the size of a 3:3 Fab/HA complex (~450 kD) were collected and used for negative stain EM imaging and reconstruction.

For EM staining and image processing, 4 μl of freshly prepared sample was incubated on a glow discharged continuous carbon 400-mesh copper grid (Electron Microscopy Sciences) for 30 s. After incubation, the grid was negatively stained with 5 × 75 μl drops of a 2% (w/v) uranyl formate solution (SPI Supplies). Excess stain solution was blotted using Watman paper and the grid was air-dried. 46B8/HA sample was analysed on a Tecnai-12 BioTween (FEI) equipped with a LaB6 filament and operated at 120 kV under low dose conditions. Images were collected using a 4 k × 4 k CCD camera (Gatan) at a nominal magnification of 62,000 × (2.22 Å per pixel). In total, 22,000 particles were manually picked from 130 CCD images and extracted using a 160px box size with the e2boxer.py software included into EMAN2 distribution[48] and subjected to reference free 2D classification with the software suite Relion[49]. In total, 22,000 particles were used to determine a 3D reconstruction of FluB HA bound to 46B8 Fab in Relion, using as a starting model the crystal structure of the B/Yamanashi/166/1998 HA (PDB ID 4M40) low pass filtered to a resolution of 60 Å[50]. This HA crystal structure and a homology model for the 46B8 Fab based on homologous antibody crystal structures and generated using the programme MOE (Chemical Computing Group) were fitted into the EM density using the fit in map algorithm in Chimera[51].

**Flow cytometry.** In total, 293 T cells were transfected with plasmids encoding the WT or mutant B/Brisbane/60/2008 or B/Victoria/504/2000 HAs or mock transfected using the calcium phosphate method. Two days later, cells were collected in the cell dissociation buffer and blocked with PBS + 5% FBS. Cells were stained with 46B8 or 34B5 in PBS (pH 7.0) + 5% FBS or a pH 4.8 buffer + 5% FBS. Cells were washed and then stained with a DyLight 649-conjugated anti-human secondary antibody (Jackson ImmunoResearch). Cells were analysed on a FACSCalibur (BD Biosciences), and results were plotted with the FlowJo 8.4.5 software. In some experiments, the mean fluorescence intensity was plotted with the Prism 6.0 software.

**IBVs and mouse studies.** All IBVs were purchased from ViraPur (San Diego). In total, 7–8-week old female DBA/2 J mice (Jackson Laboratory) were infected intranasally with 50 μl of different IBV strains diluted into influenza media at the minimum 100% lethal dose (LD$_{100}$): B/Wisconsin/1/2010 (1.5 × 10$^4$ plaque-forming units (pfu) per mouse or 3 × LD$_{50}$), B/Brisbane/60/2008 (1 × 10$^4$ pfu per mouse or 3 × LD$_{50}$), B/Victoria/504/2000 (1 × 10$^4$ pfu per mouse or 3 × LD$_{50}$), B/Russia/1/1969 (1 × 10$^5$ pfu per mouse or 9 × LD$_{50}$), B/Massachusetts/3/1966 (1 × 10$^4$ pfu per mouse or 3 × LD$_{50}$). At 24, 48 or 72 h post infection, mice received a single treatment of 46B8 or the control IgG intravenously at 15 mg kg$^{-1}$. In some experiments, mice received a single treatment of 46B8 or the control IgG intravenously at 5, 15 or 45 mg kg$^{-1}$ at 72 h post infection. For the severe mouse infection model to compare the effectiveness of Tamiflu to antibody treatment or to test the combination of Tamiflu and antibody, 7–8 weeks old female DBA/2 J mice were infected with a high lethal dose (4 × 10$^4$ pfu per mouse or 12 × LD$_{50}$) of B/Victoria/504/2000. At 72 h post infection, mice received a single treatment of 46B8 or the control IgG intravenously at 5 or 45 mg kg$^{-1}$, Tamiflu orally at 100 mg kg$^{-1}$ twice a day for 5 days, or a combined treatment of 46B8 and Tamiflu. Mice were monitored for BW and survival until 21 days after infection. For the lung titre study, 7–8 weeks old female DBA/2 J mice were infected intranasally with the high lethal dose of B/Victoria/504/2000. At 72 h post infection, mice received a single treatment of 46B8 or the control IgG intravenously at 45 mg kg$^{-1}$, Tamiflu orally at 100 mg kg$^{-1}$ twice a day for 3 days, or a combined treatment of 46B8 and Tamiflu. At day 3, 4 and 6 post infection, lung after exsanguination was collected in 2 ml influenza media and homogenized by using gentleMACS with the RNA01.01 Programme. Lung homogenates were spun at 3,000 r.p.m. for 10 min and then cleared through a 70-μm cell-strainer. Viral titres in the homogenates were determined with a TCID$_{50}$ assay on MDCK cells[52].

For in vivo fitness studies, 7–8 weeks old female DBA/2 J mice were infected with WT or mutant B/Brisbane/60/2008 viruses at 3 × 10$^4$, 1 × 10$^4$ or 5 × 10$^3$ pfu per mouse, and monitored for BW and survival. For in vivo protection studies, 7–8 weeks old female DBA/2 J mice were infected with WT or mutant B/Brisbane/60/2008 viruses at the minimum lethal dose: WT (1 × 10$^4$ pfu per mouse or 3 × LD$_{50}$), A4 (1 × 10$^4$ pfu per mouse or 3 × LD$_{50}$), B1 (5 × 10$^3$ pfu per mouse or 3 × LD$_{50}$), C2 (8 × 10$^2$ pfu per mouse or 3 × LD$_{50}$). At 72 h post-infection, mice received a single treatment of 46B8, 46B8 N297G or the control IgG intravenously at 15 mg kg$^{-1}$. In some experiments, mice received Tamiflu orally at 100 mg kg$^{-1}$ twice a day for 5 days starting from 48 h post infection. Mice were monitored for BW and survival.

All animals used in this study were housed and maintained at Genentech in accordance with American Association of Laboratory Animal Care guidelines. All experimental studies were conducted under protocols approved by the Institutional Animal Care and Use Committee of Genentech Lab Animal Research in an Association for Assessment and Accreditation of Laboratory Animal Care International-accredited facility in accordance with the Guide for the Care and Use of Laboratory Animals and applicable laws and regulations. If mice reach 30% BW loss, in addition to any of the following, they will be killed or their condition will be discussed with the veterinary staff: (a) hunched posture becomes severe; (b) when touched or encouraged to move, animal does not move from hunched position; (c) when touched or encouraged to move, animal moves, but stops and returns to

hunched position; (d) when touched or encouraged to move, animal falls over and cannot right itself; (e) when touched or encouraged to move, animal displays laboured breathing; (f) BW loss does not stabilize or increase in 48 h.

**Isolation of resistant viruses.** MDCK cells in 24-well tissue culture plates were incubated with 46B8 at 2 × IC$_{50}$ (half-maximal inhibitory concentration) in influenza media. B/Brisbane/60/2008 virus was then added at MOI 0.5 in influenza media. Every 3 or 4 days, roughly half of the supernatant was passed onto new cells and the concentration of antibody was increased 1.5-fold up to 1 × IC$_{90}$ (concentration that inhibits 90% of infection). Resistant viruses were stocked without antibody and analysed for resistance to 46B8 by plaque assay. RNAs were isolated from resistant viruses and subjected to reverse transcription, PCR amplification and sequencing of the HA gene.

**Immunogoldnegative staining EM.** WT and mutant B/Brisbane/60/2008 viruses in suspension were adsorbed to the surface of transmission electron microscope grids for 15 min. Grids were blocked with a blocking solution (Aurion Electron Microscopy Sciences) for 30 min followed by incubation with the control IgG, 34B5, 46B8 or 46B8 N297G for 1 h at room temperature. Grids were washed and incubated with a biotinylated donkey-anti-human secondary antibody (Jackson ImmunoResearch) for 1 h at room temperature. Grids were washed again and then incubated with 15 nm gold-conjugated streptavidin (Ted Pella) for 1 h at room temperature. Finally, grids were washed, treated with 4% paraformaldehyde to inactivate all virus particles and counter stained with 2% ammonium molybdate for 1 min. Samples were examined under a JEOL JEM-1400 transmission electron microscope at 120kV. Digital images were captured with a GATAN Ultrascan 1000 CCD camera at magnifications from × 1,000 to × 50,000.

**Viral genome quantification in culture supernatant.** MDCK cells were grown in 96-well tissue culture plates and infected with the WT or mutant B/Brisbane/60/ 2008 viruses at an MOI of 0.01. Culture supernatants (50 μl) were collected at day 4 post infection and subjected to RNA isolation with an SV 96 Total RNA Isolation System (Promega). cDNA synthesis was carried out with a High Capacity cDNA Reverse Transcription Kit (Applied Biosystems), followed by quantitative real-time PCR (qPCR) using a TaqMan Universal PCR Master Mix (Applied Biosystems) to detect the B/Brisbane/60/2008 M1 Matrix gene with the following oligonucleotides (5′–3′):
  Sense: GAGACACAATTGCCTACCTGCTT
  Antisense: TTCTTTCCCACCGAACCAAC
  Probe: AGAAGATGGAGAAGGCAAAGCAGAACTAGC
In some experiments, virus inoculum was removed after 2 h and cells were incubated with varying concentrations of oseltamivir acid in influenza media. Supernatants were collected at day 4 post infection and subjected to viral genome copy number determination as above. For each virus, the percent of copy numbers (normalized to the value at the lowest oseltamivir acid concentration) was plotted on the y axis versus the Log10 antibody concentrations on the x axis. The data were fit with a nonlinear regression dose–response curve with the Prism 6.0 software.

**Sequence alignment.** A multiple sequence alignment of 12,790 human IBV HA amino-acid sequences was used to assess the genetic diversity of potential 46B8-contacting residues. The alignment was visualized in JalView. Sequence analysis was performed using the SNP tool provided via the Influenza Research Database website, which computes the alignment with the MUSCLE tool. The data were used to generate a frequency table of amino acids observed at each individual position.

**Whole-genome sequencing.** RNAs were extracted from the B/Brisbane/60/2008 WT and resistant viruses and subjected to one-step reverse transcription-PCR amplification using the published universal IBV-GA2 primer cocktail and temperature cycle parameters[53] with the SuperScript III system (Life Technologies). A fraction of the PCR products were resolved and visualized on 1% agarose gel to ensure the presence of all eight influenza B genome segments at the expected sizes. The remaining PCR products were purified using the QIAquick PCR purification kit (Qiagen) and subjected to next-generation sequencing on the Illumina platform. Libraries were generated using the TruSeq Nano DNA Library Preparation Kit (Illumine) on the automated platform NeoPrep (Illumina). Sizes of the libraries were confirmed using Fragment Analyzer (Advanced Analytical Technologies) and the concentrations were determined by Qubit (Thermo Fisher Scientific). The libraries were multiplexed and sequenced on Illumina HiSeq (Illumina) to generate ∼60M of paired end 75 base pair reads per library. Reads of the WT and mutant viruses were mapped onto the B/Brisbane/60/ 2008 reference genome (GenBank Accession Numbers KC866603, KC866604, KC866602, KC866605, FJ766839, KC866607 and KC866606) using GSNAP[54]. Variant detection was carried out using in-house R scripts, which utilize the Bioconductor packages, GenomicRanges[55], Genomic Alignments[55], VariantTools[56] and gmapR[57]. Only base-calls with Q-score ≥ 30 were tallied for variant calling. Amino-acid substitutions with respect to the reference sequences were determined at a frequency threshold of 50%.

**Statistical analyses.** Statistical analysis was performed with the Prism 6.0 software. An unpaired two-tailed *t*-test with 95% confidence interval was used for calculation of *P* values. For mouse survival experiments, a log-rank (Mantel–Cox) test was used for calculation of *P* values. Group sizes and *P* values for each experiment are given in figure legends. *P* values < 0.05 were considered significant.

**Data availability.** The data that support the findings of this study are available from the corresponding authors upon request.

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

## Acknowledgements

We would like to thank Eric Brown (Genentech), Sharookh Kapadia (Genentech) and Elizabeth Newton (Genentech) for careful review of the manuscript. We would also like to

thank Olga Li (Genentech) for advice on next-generation sequencing, and Janina Reeder (Genentech) and Alexander Abbas (Genentech) for help with the sequence alignment.

## Author contributions

N.C., L.R.S. and M.-W.T. designed the research; N.C., S.P., G.N., N.C., A.E., R.F., L.K., E.K., M.R., Z.L., H.C., Z.M., J.S. and C.C. performed the experiments; N.C., L.R.S., L.K., E.S., M.X., P.L. and C.C. analysed the data; N.C. and M.-W.T. wrote the paper.

## Additional information

**Competing financial interests:** N.C., S.P., G.N., N.C., A.E., R.F., L.K., E.K., M.R., Z.L., H.C., E.S., Z.M., J.S., M.X., P.L., C.C. and M.-W.T. are employees of Genentech; L.R.S. is an employee of Achaogen.

DOI: 10.1038/ncomms15779    **OPEN**

# Corrigendum: A broadly protective therapeutic antibody against influenza B virus with two mechanisms of action

Ning Chai, Lee R. Swem, Summer Park, Gerald Nakamura, Nancy Chiang, Alberto Estevez, Rina Fong, Lynn Kamen, Elviza Kho, Mike Reichelt, Zhonghua Lin, Henry Chiu, Elizabeth Skippington, Zora Modrusan, Jeremy Stinson, Min Xu, Patrick Lupardus, Claudio Ciferri & Man-Wah Tan

*Nature Communications* 8:14234 doi: 10.1038/ncomms14234 (2017); Published 19 Jan 2017; Updated 25 May 2017

This Article contains errors in Fig. 5c. The blue squares should be labelled 'Water + 46B8' and the green triangles should be labelled 'Tamiflu + 46B8'. The correct version of Fig. 5 appears below as Fig. 1.

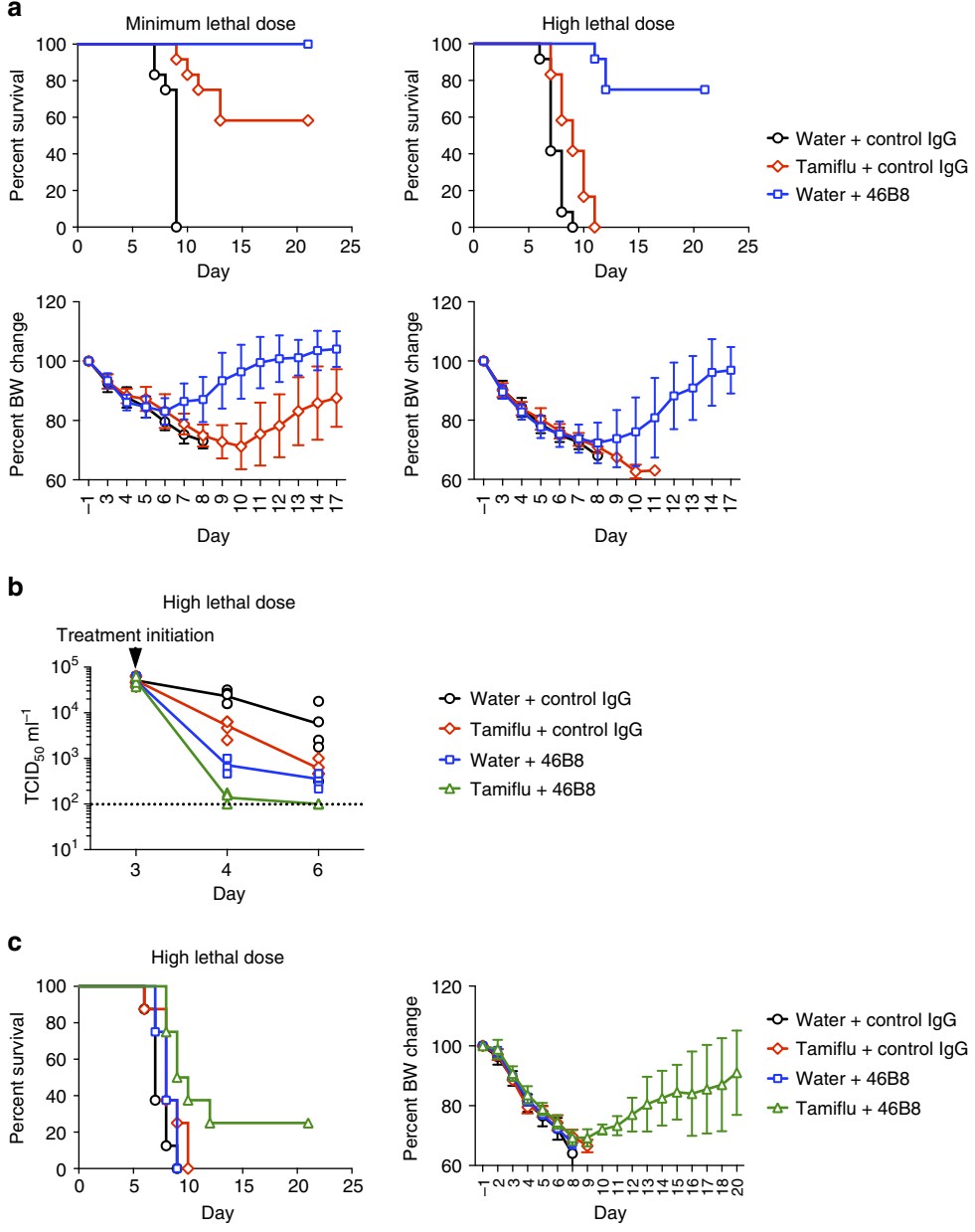

**Figure 1 |**

