## [Peer Review File · Nature Communications]

Reviewers' comments:

Reviewer #1 (Remarks to the Author):

In their manuscript Chai and colleagues describe a broadly neutralizing anti-HA head antibody that recognizes all tested influenza B HAs. The mAb binds to the side of the HA head and the authors speculate that it works through inhibition of fusion as well as through ADCC. The mAb is certainly interesting but there are several open questions that need the authors' attention.

Major points

1) The number of tested influenza B strains is low and no 2015/16 isolates were tested. It is uncertain if the binding of the mAb is really covering all influenza B strains.

2) The authors claim that this is the first influenza B antibody to use ADCC as mechanism. However, this has also been described for CR9114 (which actually doesn't neutralize influenza B but protects mice from infection). Also, 46B8 is not the first antibody that binds to the esterase domain and has ADCC activity. A recent paper by Dilillo et al. in JCI describes several neutralizing, non-HI mAbs for influenza A that depend on Fc-FcR interactions.

3) Throughout the manuscript the authors make comparisons to data published for mAb CR8071 and to a lesser degree to mAb 5A7. The assays used to test these mAbs are different and if the author's want to claim that these mAbs use different mechanisms than mAb 46B8 they need to test them side by side. The CDRs of both mAbs have been published in patents and can easily be generated.

4) Extreme mAb doses (45 mg/kg!!!) are needed for optimal efficacy. As comparison, mAb CR8071 protects at a dose as low as 1.7 mg/kg.

5) The authors claim that CR8071 does not protect therapeutically. This is unknown since it was not tested therapeutically so far. The authors should do a head to head comparison or they should drop that claim (page 18, line 9).

6) The authors claim that mAb 46B8 works through neutralization and ADCC. The data on this is unclear. They should test:

a) the Fc-mutant of 46B8 in the mouse model to gauge the importance of the Fc-mediated mechanism

b) if the mAb really works through ADCC or if it works through another Fc-FcR mediated mechanism

7) The challenge dose is a little unclear. Please state the challenge dose in LD50 to make it comparable to previous publications.

8) It is unclear what criteria was used as humane endpoint for the mouse experiment. In

any case by looking at S. Figures 9 and 10 it becomes clear that some of the groups lost more than 30% of their initial body weight. This is highly unethical. Standard humane endpoints in the US are between 20-25% weight loss, most European IACUCs mandate 10-20%. What was the humane endpoint (death is not a humane endpoint) as approved by the IACUC? What rationale was given to the IACUC to get an endpoint below 30% body weight loss approved? It would be good to add a statement from the IACUC to indicate why they think that this is still in accordance with applicable laws and regulations.

9) For figures 4 and 5 the weight loss curves should be shown in the main manuscript as well. This is specifically important for figure 5. With the weight loss data it becomes clear that under regular circumstances (e.g. with a 20% weight loss endpoint) the mAb would not be efficacious at all. Also, the standard deviation is missing.

10) Figure 7: The y-axis of the weight loss should show the range from 60-120 so that the reader can actually see the differences.

Minor points

1) Page 3, line 4: 'flu' is not a scientific term

2) Page 8: The authors use antibody 34B5 to show that low pH and DTT treatment remove the HA1 domain from HA transfected cells. How can the authors be sure that the mAb is stable under these conditions? Maybe it just falls apart, can't bind anymore but the HA1 is still on the cells?

3) Was the full virus sequenced for the escape mutants or just the HA? Where there mutations in other genomic segments?

4) Page 24, line 21: This is a multicycle replication neutralization assay, not an entry assay.

5) Page 28, line 10-11: It is unclear how the normalization was done. Please explain and give a rationale why the data was normalized to the lowest antibody concentration and not to irrelevant mAb?

6) Page 29, line 17: Please specify the composition of the dissociation buffer.

7) S. Fig. 9 and S. Fig. 10 lack the error bars. Standard deviation should be shown.

Reviewer #2 (Remarks to the Author):

The manuscript describes the isolation and characterization of flu B neutralizing antibodies isolated from immunized patients, in particular 46B8, a pan-flu B antibody with a dual mechanism of action. Overall, this is a very thorough study of an interesting human antibody that targets a conserved, yet underappreciated epitope on the influenza surface

protein hemagglutinin. The most remarkable features of this antibody are the breadth of neutralization in flu B that spans ~50 years of evolution, and the relatively potent protection in mouse models relative to antibodies that target a similar epitope. In that regard the antibody appears to span two protomers of HA and blocks low pH induced conformational changes in HA (mechanism 1). Of potential concern is the relative ease of escape with a single point mutation in residue S301 during serial passage without any change in viral fitness. Somewhat paradoxically however is that S301 is 99% conserved in flu B. Interestingly, this mutation and others rationally made using structural data appeared to influence the neutralization in a pH dependent manner. Thus, by inference, escape was not mediated at the cell surface, but rather after being taken up into the endosome. Therefore, the antibody was still effective in vivo at protecting via ADCC (mechanism 2) but not CDC, as shown by the authors. Finally, there is data that shows 46B8 synergy with the neuraminidase inhibitor, Tamiflu, and maybe potential for a dual therapeutic approach to protect against flu B.

This was an enjoyable read that incorporates many assays and analytics, that support the conclusions drawn in the paper and should be accepted with minor revisions.

Specific points:

I found the following statement awkward because the mutations that they chose were based on a rational attempt to disrupt binding, not naturally occurring mutations: "Importantly, the mutations that reduced 46B8 binding are either non-existing or present at very low frequency among IBV field isolates (Fig. 3d, bottom panel).

Better delineation of interprotomer binding in Fig. 3a. Protomers could be colored differently, for example.

The Discussion is very redundant with the Results and could be cut way down.

Reviewer #3 (Remarks to the Author):

Chai et al generate a pan-IAB MAb and show that it is effective in murine Influenza infection experiments and that ADCC may play a role in infection. The work is well done and should be of interest to Nature Communication readers.

Comments

It wasn't clear in the MS the Fc backbone of the 46B8 mAb - I gather it was cloned out - was it placed then on a standard IgG1 Fc or was the Fc region native to the original plasmablast. The mAb is still useful either way, but the issue goes to what was isolated and the potential Fc-mediated function(s) of the native Ab.

There appears to be only partial resistance to 46B8 in the A4, B1, and C2 mutants, with

significant binding at pH7. The partial resistance could explain survival, albeit with some enhanced weight loss, of mice treated with 46B8 and infected with the mutants, rather than being explained by ADCC as proposed. I can foresee some problems with dosing and interpretation, but did the authors consider treating mice with the N297G 46B8 mutant Ab then infecting with the resistant viruses (perhaps at a dose of virus or MAb where differences could be amplified)?

Reviewers' comments:

Reviewer #1 (Remarks to the Author):

In their manuscript Chai and colleagues describe a broadly neutralizing anti-HA head antibody that recognizes all tested influenza B HAs. The mAb binds to the side of the HA head and the authors speculate that it works through inhibition of fusion as well as through ADCC. The mAb is certainly interesting but there are several open questions that need the authors' attention.

Major points

1) The number of tested influenza B strains is low and no 2015/16 isolates were tested. It is uncertain if the binding of the mAb is really covering all influenza B strains.

Since 2015/16 isolates are not available commercially, we tested two additional influenza B strains that were used in most recent influenza vaccines: B/Phuket/3073/2013 and B/Massachusetts/2/2012. 46B8 neutralized these two viruses with IC₅₀S comparable to the earlier strains.

We have now added the results to Table 1 (Page 55).

2) The authors claim that this is the first influenza B antibody to use ADCC as mechanism. However, this has also been described for CR9114 (which actually doesn't neutralize influenza B but protects mice from infection). Also, 46B8 is not the first antibody that binds to the esterase domain and has ADCC activity. A recent paper by Dilillo et al. in JCI describes several neutralizing, non-HI mAbs for influenza A that depend on Fc-FcR interactions.

We thank the referee for the careful review. We made these claims for the following reasons. (a) Although one can infer from the Dreyfus et. al. 2012 paper that CR9114 possibly protects mice from influenza B infection via ADCC, it was not directly tested. The observed in vivo protection could be mediated, among others by ADCC or complement-dependent cytotoxicity (CDC). By contrast, we directly showed that 46B8 uses ADCC, but not CDC as a mechanism. (b) We checked the list of mAbs described in the Dilillo et al. 2016 paper and none of them has been demonstrated to bind to the esterase domain although the binding sites of some of the mAbs are not known. To be cautious, we have now removed the sentence "To our knowledge, this is the first report of an esterase domain binding mAb that is able to induce ADCC".

Please refer to Discussion (Page 24 Line 18 – 19).

3) Throughout the manuscript the authors make comparisons to data published for mAb CR8071 and to a lesser degree to mAb 5A7. The assays used to test these mAbs are different and if the author's want to claim that these mAbs use different mechanisms than mAb 46B8 they need to test them side by side. The CDRs of both mAbs have been published in patents and can easily be generated.

We agree with the referee that without side-by-side studies, we should not claim that these mAbs use different mechanisms. We have now removed all the comparisons between these mAbs.

Please refer to Results (Page 10 Line 13 – 15 and 17 – 19) and Discussion (Page 21 Line 5 – 20 and Page 22 Line 10 – 13).

4) Extreme mAb doses (45 mg/kg!!!) are needed for optimal efficacy. As comparison, mAb CR8071 protects at a dose as low as 1.7 mg/kg.

In most of our mouse studies, 15 mg/kg of 46B8 achieved optimal efficacy. We also showed in some experiments (Figure 4b) that 5 mg/kg of 46B8 was partially protective. We agree with the referee that without side-by-side studies, we should not claim that these mAbs (46B8, CR8071 and 5A7) have different efficacies. We have now removed all the comparisons between these mAbs.

Please refer to Introduction (Page 5 Line 14), Results (Page 10 Line 17 – 19) and Discussion (Page 20 Line 6 – 13).

5) The authors claim that CR8071 does not protect therapeutically. This is unknown since it was not tested therapeutically so far. The authors should do a head to head comparison or they should drop that claim (page 18, line 9).

We agree with the referee that since the efficacy of CR8071 was demonstrated in a prophylactic setting but was not tested therapeutically, and we did not perform a head to head comparison, to more accurately reflect this we have rephrased the sentence to "... and protected mice from lethal infection when administered prophylactically" in Introduction and removed the claim in Discussion.

Please refer to Introduction (Page 5 Line 1 – 2) and Discussion (Page 20 Line 9 – 10).

6) The authors claim that mAb 46B8 works through neutralization and ADCC. The data on this is unclear. They should test:

a) the Fc-mutant of 46B8 in the mouse model to gauge the importance of the Fc-mediated mechanism

Following the referee's advice, we tested the Fc-mutant mAb 46B8 N297G in the mouse model against the WT or mutant B/Brisbane/60/2008 viruses (Figure 9b and 9c). 46B8 N297G fully protected mice infected with the WT virus but completely lost efficacy against the A4 mutant, supporting the importance of Fc functions in in vivo protection. Interestingly, 90% of the mice infected with the B1 or C2 mutant were protected by 46B8 N297G, possibly due to partial sensitivity of these two viruses to the mAb observed in a neutralization assay measuring viral nucleoprotein (NP) expression at 16 hr post-infection (Supplementary Figure 10). However, we do not know if this partial sensitivity could fully explain the difference in mouse protection because the initial 4-day plaque reduction assay gave slightly different results as mutant B1 appeared to be the most resistant in the plaque assay (Figure 6a). In general, plaque reduction assay is more sensitive and requires less amount of antibody to achieve neutralization. We do not know the correlation between the antibody concentrations (ng/ml) used for the in vitro assays and the doses (mg/kg) used for the mouse studies.

Because of this discrepancy, we explored other factors that might contribute to the differential sensitivities of A4 and B1/C2 to 46B8 N297G in vivo. First, we tested binding of 46B8 N297G to the mutant virus particles by immunogold EM and did not see difference (Supplementary Figure 11). Second, we did whole genome sequencing of the viruses to explore mutations in other viral proteins that might cause the differential in vivo sensitivities (Supplementary Table 1). We identified three mutations that are unique to A4 (R185K) or B1/C2 (I303M and E582G) in the PB2 protein. These substitutions were not reported previously but could affect viral replication.

We have now added Figure 9b and 9c and Supplementary Figure 10 and 11 and Supplementary Table 1. Please also refer to Results (Page 14 Line 21 – 23, Page 15 Line 19 – 22 and Page 18 Line 2 – Page 19 Line 2) and Discussion (Page 24 Line 19 – 22).

b) if the mAb really works through ADCC or if it works through another Fc-FcR mediated mechanism

We showed that 46B8 works through ADCC but not CDC.

Please refer to Figure 2 and Supplementary Figure 6.

7) The challenge dose is a little unclear. Please state the challenge dose in LD50 to make it comparable to previous publications.

We have added the challenge doses in LD₅₀. It is now made explicit in Methods.

Please refer to Methods (Page 36 Line 19 – 22, Page 37 Line 5 and Page 37 Line 23 – Page 38 Line 2).

8) It is unclear what criteria was used as humane endpoint for the mouse experiment. In any case by looking at S. Figures 9 and 10 it becomes clear that some of the groups lost more than 30% of their initial body weight. This is highly unethical. Standard humane endpoints in the US are between 20-25% weight loss, most European IACUCs mandate 10-20%. What was the humane endpoint (death is not a humane endpoint) as approved by the IACUC? What rationale was given to the IACUC to get an endpoint below 30% body weight loss approved? It would be good to add a statement from the IACUC to indicate why they think that this is still in accordance with applicable laws and regulations.

All our mouse study protocols were approved by IACUC. Below are the monitoring criteria:

Starting day 5 of infection, all mice are monitored twice a day and at least 5 hours apart between observations. Mice with a body condition score (BCS) <2 will be very closely monitored and/or the Veterinary Staff will be contacted (See BCS guidelines on the LAR/IACUC homepage). During observations, mice will be monitored for clinical appearance, including body condition, coat appearance, posture, lethargy, cyanotic paws and tails. Moribund animals displaying severe effects will be immediately euthanized or discussed with the veterinary staff immediately. Starting 14 days after infection, animals will be monitored at least 3 times a week, with increasing frequency if needed due to adverse effects, including mortality of other animals on study. Monitoring frequency will be increased up to twice daily (5 hours apart) as directed by the veterinary staff. Mice will be weighed twice a week throughout the study. Mice losing 15% body weight will be weighed daily. Daily weights will be provided to the veterinary staff. If mice reach 30% body weight loss, in addition to any of the following, they will be euthanized or their condition will be discussed with the veterinary staff.

1. Hunched posture becomes severe.
2. When touched or encouraged to move, animal does not move from hunched position.
3. When touched or encouraged to move, animal moves, but stops and returns to hunched position.
4. When touched or encouraged to move, animal falls over and cannot right itself.
5. When touched or encouraged to move, animal displays labored breathing.
6. Body weight loss does not stabilize or increase in 48 hours.

Following the referee's advice, we have now added a statement from the IACUC regarding 30% body weight loss. Please refer to Methods (Page 38 Line 12 – 20).

9) For figures 4 and 5 the weight loss curves should be shown in the main manuscript as well. This is specifically important for figure 5. With the weight loss data it becomes clear that under regular circumstances (e.g. with a 20% weight loss endpoint) the mAb would not be efficacious at all.

Also, the standard deviation is missing.

Following the referee's advice, we have now added standard deviation to all our body weight data. We have also moved the body weight data of Figure 5 to the main manuscript. We have kept the body weight data of Figure 4 in the Supplementary Information due to space limit of the main manuscript.

Please refer to Figure 5, 7 and 9 and Supplementary Figure 9 and 12.

10) Figure 7: The y-axis of the weight loss should show the range from 60-120 so that the reader can actually see the differences.

Following the referee's advice, we have now changed the y-axis range to 60-120.

Please refer to Figure 7.

Minor points

1) Page 3, line 4: 'flu' is not a scientific term

We thank the referee for the careful review. We have now changed it to "influenza".

Please refer to Introduction (Page 3 Line 4).

2) Page 8: The authors use antibody 34B5 to show that low pH and DTT treatment remove the HA1 domain from HA transfected cells. How can the authors be sure that the mAb is stable under these conditions? Maybe it just falls apart, can't bind anymore but the HA1 is still on the cells?

We showed that when cells were pre-incubated with 46B8, 34B5 was able to bind even after low pH and DTT treatment, indicating that 34B5 was stable under low pH and DTT conditions and did not fall apart. In cells pre-incubated with the control IgG and treated with low pH, 34B5 was still able to bind without DTT treatment; it lost binding only after DTT treatment, indicating dissociation of HA1 upon disulfide bond breakage. This is an established assay used by Ekiert et al. (Science 333:843, 2011) and Dreyfus et al. (Science 337: 1343, 2012) to demonstrate prevention of low pH-induced HA conformational changes by mAb CR8020 and CR9114.

3) Was the full virus sequenced for the escape mutants or just the HA? Where there mutations in other genomic segments?

Yes. Following the referee's suggestion, we did whole genome sequencing of the viruses to explore mutations in other genomic segments (Supplementary Table 1). We identified three mutations unique to A4 (R185K) or B1/C2 (I303M and

E582G) in the PB2 protein that might contribute to their differential sensitivities in vivo. These substitutions were not reported previously but might affect viral replication. We also found two mutations (K338R and V625A) in the PA protein. Since they are not unique to A4 or B1/C2, they are less likely to contribute to the differential in vivo sensitivities of the mutant viruses. A few other mutations with lower frequencies were identified in the PB1, PB2, PA, NA, BM2 and NS1 proteins. Their effects on viral fitness are not known.

We have now added Supplementary Table 1. Please also refer to Results (Page 18 Line 14 – Page 19 Line 2).

4) Page 24, line 21: This is a multicycle replication neutralization assay, not an entry assay.

We have now changed it to “neutralization assay”. Please refer to Methods (Page 27 Line 4).

5) Page 28, line 10-11: It is unclear how the normalization was done. Please explain and give a rationale why the data was normalized to the lowest antibody concentration and not to irrelevant mAb?

As we mentioned in Methods (Page 27 Line 16 – 18 and Page 30 Line 21 – 23), infection of each virus was normalized to the number of infected cells at the lowest antibody concentration. Percent of infection was plotted against the antibody concentrations. The normalization was done with the Prism 6 software. Consequently, the neutralization curves for all viruses start with 100% infection at the lowest antibody concentration and are easy to compare between viruses. It also avoids the problem of plotting 0 (antibody concentration) on a log scale. Importantly, 0% infection is defined as 0 infected cells (not the number of infected cells at the highest antibody concentration) so that partial neutralization can be truly reflected. This is the way normalization is done in all our neutralization assays, and it is commonly used in literature. When we determine the range of antibody concentrations, we always make sure that infection saturates at the lowest antibody concentration, meaning that the number of infected cells at the lowest antibody concentration is similar to the number in the presence of an irrelevant antibody.

6) Page 29, line 17: Please specify the composition of the dissociation buffer.

We used a commercial dissociation buffer. We have now added the name of the commercial supplier.

Please refer to Methods (Page 32 Line 6).

7) S. Fig. 9 and S. Fig. 10 lack the error bars. Standard deviation should be

shown.

Following the referee's advice, we have now added standard deviation to all our body weight data.

Please refer to Figure 5, 7 and 9 and Supplementary Figure 9 and 12.

Reviewer #2 (Remarks to the Author):

The manuscript describes the isolation and characterization of flu B neutralizing antibodies isolated from immunized patients, in particular 46B8, a pan-flu B antibody with a dual mechanism of action. Overall, this is a very thorough study of an interesting human antibody that targets a conserved, yet underappreciated epitope on the influenza surface protein hemagglutinin. The most remarkable features of this antibody are the breadth of neutralization in flu B that spans ~50 years of evolution, and the relatively potent protection in mouse models relative to antibodies that target a similar epitope. In that regard the antibody appears to span two protomers of HA and blocks low pH induced conformational changes in HA (mechanism 1). Of potential concern is the relative ease of escape with a single point mutation in residue S301 during serial passage without any change in viral fitness. Somewhat paradoxically however is that S301 is 99% conserved in flu B. Interestingly, this mutation and others rationally made using structural data appeared to influence the neutralization in a pH dependent manner. Thus, by inference, escape was not mediated at the cell surface, but rather after being taken up into the endosome. Therefore, the antibody was still effective in vivo at protecting via ADCC (mechanism 2) but not CDC, as shown by the authors. Finally, there is data that shows 46B8 synergy with the neuraminidase inhibitor, Tamiflu, and maybe potential for a dual therapeutic approach to protect against flu B.

This was an enjoyable read that incorporates many assays and analytics, that support the conclusions drawn in the paper and should be accepted with minor revisions.

Specific points:

I found the following statement awkward because the mutations that they chose were based on a rational attempt to disrupt binding, not naturally occurring mutations: "Importantly, the mutations that reduced 46B8 binding are either non-existing or present at very low frequency among IBV field isolates (Fig. 3d, bottom panel).

Better delineation of interprotomer binding in Fig. 3a. Protomers could be

colored differently, for example.

The Discussion is very redundant with the Results and could be cut way down.

Response to the Specific points:

1) I found the following statement awkward because the mutations that they chose were based on a rational attempt to disrupt binding, not naturally occurring mutations: "Importantly, the mutations that reduced 46B8 binding are either non-existing or present at very low frequency among IBV field isolates (Fig. 3d, bottom panel).

We agree with the referee's point. We have now removed this statement and the bottom panel of Figure 3d.

Please refer to Figure 3d and Results (Page 11 Line 16 – 18).

2) Better delineation of interprotomer binding in Fig. 3a. Protomers could be colored differently, for example.

Following the referee's advice, we have now colored the three protomers with different shades of blue.

Please refer to Figure 3a.

3) The Discussion is very redundant with the Results and could be cut way down.

Following the referee's advice, we have now removed the content in Discussion that is redundant with the Introduction or Results.

Please refer to Discussion (Page 20 Line 2 – 18 and 20 – 21, Page 21 Line 5 – 20, Page 23 Line 19 – 21 and Page 24 Line 16 – 18).

Reviewer #3 (Remarks to the Author):

Chai et al generate a pan-IAB MAb and show that it is effective in murine Influenza infection experiments and that ADCC may play a role in infection. The work is well done and should be of interest to Nature Communication readers.

Comments

It wasn't clear in the MS the Fc backbone of the 46B8 mAb - I gather it was

cloned out - was it placed then on a standard IgG1 Fc or was the Fc region native to the original plasmablast. The mAb is still useful either way, but the issue goes to what was isolated and the potential Fc-mediated function(s) of the native Ab.

There appears to be only partial resistance to 46B8 in the A4, B1, and C2 mutants, with significant binding at pH7. The partial resistance could explain survival, albeit with some enhanced weight loss, of mice treated with 46B8 and infected with the mutants, rather than being explained by ADCC as proposed. I can foresee some problems with dosing and interpretation, but did the authors consider treating mice with the N297G 46B8 mutant Ab then infecting with the resistant viruses (perhaps at a dose of virus or MAb where differences could be amplified)?

Reponse to the comments:

1) It wasn't clear in the MS the Fc backbone of the 46B8 mAb - I gather it was cloned out - was it placed then on a standard IgG1 Fc or was the Fc region native to the original plasmablast. The mAb is still useful either way, but the issue goes to what was isolated and the potential Fc-mediated function(s) of the native Ab.

We thank the referee for pointing out this lack of description. We cloned out the variable regions of the heavy and light chains from the sorted plasmablasts and inserted them into mammalian expression vectors containing standard human IgG1 heavy and light chain constant regions. Consequently, the Fc region of the resulting mAbs is not native to the original plasmablasts. We have now added this description to Methods.

Please refer to Methods (Page 26 Line 11 – 16).

2) There appears to be only partial resistance to 46B8 in the A4, B1, and C2 mutants, with significant binding at pH7. The partial resistance could explain survival, albeit with some enhanced weight loss, of mice treated with 46B8 and infected with the mutants, rather than being explained by ADCC as proposed. I can foresee some problems with dosing and interpretation, but did the authors consider treating mice with the N297G 46B8 mutant Ab then infecting with the resistant viruses (perhaps at a dose of virus or MAb where differences could be amplified)?

We thank the referee for the advice. We used two in vitro assays to test the sensitivities of the three mutant viruses to 46B8. The results of the two assays were slightly different: in a 4-day plaque reduction assay (Figure 6a), B1 appeared to be the most resistant; in a 16-hr infection neutralization assay measuring viral nucleoprotein (NP) expression (Supplementary Figure 10), A4 was more resistant than B1/C2.

Following the referee's suggestion, we tested the Fc-mutant mAb 46B8 N297G in the mouse model against the WT or mutant B/Brisbane/60/2008 viruses (Figure 9b and 9c). 46B8 N297G fully protected mice infected with the WT virus but completely lost efficacy against the A4 mutant, supporting the importance of Fc functions in in vivo protection. Interestingly, 90% of the mice infected with the B1 or C2 mutant were protected by 46B8 N297G, possibly due to the partial sensitivity of these two viruses to the mAb observed in the infection neutralization assay (Supplementary Figure 10). However, we do not know if this partial sensitivity could fully explain the difference in mouse protection because the 4-day plaque reduction assay gave slightly different results (Figure 6a). In general, plaque reduction assay is more sensitive and requires less amount of antibody to achieve neutralization. We do not know the correlation between the antibody concentrations (ng/ml) used for the in vitro assays and the doses (mg/kg) used for the mouse studies.

Because of this discrepancy, we explored other factors that might contribute to the differential sensitivities of A4 and B1/C2 to 46B8 N297G in vivo. First, we tested binding of 46B8 N297G to the mutant virus particles by immunogold EM and did not see difference (Supplementary Figure 11). Second, we did whole genome sequencing of the viruses to explore mutations in other viral proteins that might cause the differential in vivo sensitivities (Supplementary Table 1). We identified three mutations that are unique to A4 (R185K) or B1/C2 (I303M and E582G) in the PB2 protein. These substitutions were not reported previously but could affect viral replication.

We have now added Figure 9b and 9c and Supplementary Figure 10 and 11 and Supplementary Table 1. Please also refer to Results (Page 14 Line 21 – 23, Page 15 Line 19 – 22 and Page 18 Line 2 – Page 19 Line 2) and Discussion (Page 24 Line 19 – 22).

Reviewers' comments:

Reviewer #1 (Remarks to the Author):

Re-review Chai et al.:

The author's addressed most points but there are still some questions that remain open:

Reviewer 1/Major point 2: Crucell just published the data:

<http://journal.frontiersin.org/article/10.3389/fimmu.2016.00399/full>. The authors should read the paper and revise their response accordingly.

Reviewer 1/Major point 6:

a) The data/response is not really clear, specifically regarding the role of the mutation in the PB2 protein.

b) This referred to cellular mechanism including ADCP, ADRB, antibody-dependent netosis etc.. It is unclear and unlikely that ADCC is the protective mechanism (unless it is shown), e.g. by a NK depletion experiment prior to challenge.

Reviewer 1/Minor point 3: Mutations in other proteins than the HA might impact on viral fitness and skew the results. The good way to solve this is to clone the 'escaped' HA and rescue it in a clean backbone.

Reviewer #2 (Remarks to the Author):

The authors have satisfactorily addressed the reviewer concerns.

Reviewer #3 (Remarks to the Author):

I am satisfied with the response.

Reviewers' comments:

Reviewer #1 (Remarks to the Author):

Re-review Chai et al.:

The author's addressed most points but there are still some questions that remain open:

Major point 2: Crucell just published the data:

<http://journal.frontiersin.org/article/10.3389/fimmu.2016.00399/full>. The authors should read the paper and revise their response accordingly.

We thank the referee for bringing up this recent study. We have read the paper and discussed it in the revised manuscript.

Please refer to Introduction (Page 5 Line 1) and Discussion (Page 22 Line 10 – 13).

Major point 6:

a) The data/response is not really clear, specifically regarding the role of the mutation in the PB2 protein.

Regarding the effects of PB2 mutations on viral fitness and in vivo sensitivity to mAb, we have demonstrated in Figure 7 that the three mutant viruses do not show reduced fitness in vitro or in vivo, indicating that the PB2 mutations are not likely to impair viral replication or virulence in mouse. We have now discussed this point in the Results section in the revised manuscript to make it clear.

Please refer to Results (Page 18 Line 5 – 8).

b) This referred to cellular mechanism including ADCP, ADRB, antibody-dependent netosis etc.. It is unclear and unlikely that ADCC is the protective mechanism (unless it is shown), e.g. by a NK depletion experiment prior to challenge.

We have added discussion of other Fc-mediated cellular mechanisms that could contribute to protection in vivo in the Discussion section of the revised manuscript. In addition, we have generated more representative data from repeated in vivo protection experiments and modified Figure 9. While 46B8 N297G (the Fc mutant mAb) fully protected against the WT virus, it lost efficacy against the mutant viruses to different degrees. Mutant A4 was largely resistant to 46B8 N297G whereas B1 and C2 showed partial sensitivities, consistent with their partial sensitivities to the mAb in the in vitro neutralization assay (Supplementary Figure 10). These results clearly showed the importance of Fc functions in protection in vivo. We have also modified the Results and Discussion sections accordingly.

Please refer to Figure 9, Results (Page 17 Line 17) and Discussion (Page 21 Line 23 – Page 22 Line 6).

Minor point 3:

Mutations in other proteins than the HA might impact on viral fitness and skew the results. The good way to solve this is to clone the 'escaped' HA and rescue it in a clean backbone.

We agree that the non-HA mutations identified via whole genome sequencing are interesting and worth further studies, but we feel that these studies are beyond the scope of this manuscript. We have benchmarked our manuscript to the scope of published studies on broadly neutralizing anti-influenza mAbs. These publications, including the first report of a pan-IAV mAb FI6 (Science 333:850, 2011) and the most recent one of another pan-IAV mAb in Cell 166:596, 2016, did not describe the selection of escape viruses. In the few studies reporting escape viruses, none of them characterized the viruses in depth (e.g. viral fitness, sensitivity to mAb in mouse and whole genome sequencing), including the study describing the three anti-IBV mAbs (Science 337:1343, 2012). Compared to the literature, the data presented in our manuscript cover all important aspects of a broadly neutralizing mAb, including antibody discovery, in vitro and in vivo efficacies, mechanisms of action, structural data and epitope mapping, resistant virus isolation and in-depth characterization. That notwithstanding, we understand the value of rescuing the "escaped" HA in a clean backbone for future studies and had therefore attempted to obtain a reverse genetics system from St. Jude, the originator of this reagent. We were however unsuccessful due in part to contractual reasons.

Reviewer #2 (Remarks to the Author):

The authors have satisfactorily addressed the reviewer concerns.

Reviewer #3 (Remarks to the Author):

I am satisfied with the response.

REVIEWERS' COMMENTS:

Reviewer #1 (Remarks to the Author):

I think the authors addressed most points except the last one from the re-review. There are excellent anti-HA mAb papers out there that in detail characterize escape mutants (e.g. <https://www.ncbi.nlm.nih.gov/pubmed/27281570>). Also, in my opinion science is not about 'benchmarking what other people did'.

REVIEWERS' COMMENTS:

Reviewer #1 (Remarks to the Author):

I think the authors addressed most points except the last one from the re-review. There are excellent anti-HA mAb papers out there that in detail characterize escape mutants (e.g. <https://www.ncbi.nlm.nih.gov/pubmed/27281570>). Also, in my opinion science is not about 'benchmarking what other people did'.

We agree with the referee that science is not about benchmarking what other people did. We also agree with the referee (from previous reviews) that side-by-side studies are needed in order to compare our results with others, and that the non-HA mutations we identified via whole genome sequencing are interesting and worth further studies. However, we feel that those studies are beyond the scope of this manuscript. We believe the results presented in the current manuscript cover all the important aspects of a novel broadly neutralizing mAb and are sufficient for publication in a prestigious journal like Nature Communications.